# Reinforcement Learning from Bagged Reward

**Yuting Tang**[*]  *tang@ms.k.u-tokyo.ac.jp*
*The University of Tokyo & RIKEN Center for AIP*
*Tokyo, Japan*

**Xin-Qiang Cai**[*]  *xinqiang.cai@riken.jp*
*RIKEN Center for AIP*
*Tokyo, Japan*

**Yao-Xiang Ding**  *dingyx.gm@gmail.com*
*State Key Lab for CAD & CG, Zhejiang University*
*Hangzhou, China*

**Qiyu Wu**  *qiyuw@logos.t.u-tokyo.ac.jp*
*The University of Tokyo*
*Tokyo, Japan*

**Guoqing Liu**  *guoqingliu@microsoft.com*
*Microsoft Research AI4Science*
*Beijing, China*

**Masashi Sugiyama**  *sugi@k.u-tokyo.ac.jp*
*RIKEN Center for AIP & The University of Tokyo*
*Tokyo, Japan*

**Reviewed on OpenReview:** *https://openreview.net/forum?id=bXUipBbZDA*

## Abstract

In Reinforcement Learning (RL), it is commonly assumed that an immediate reward signal is generated for each action taken by the agent, helping the agent maximize cumulative rewards to obtain the optimal policy. However, in many real-world scenarios, designing immediate reward signals is difficult; instead, agents receive a single reward that is contingent upon a partial sequence or a complete trajectory. In this work, we define this challenging problem as *RL from Bagged Reward* (RLBR), where sequences of data are treated as *bags* with non-Markovian bagged rewards, leading to the formulation of Bagged Reward Markov Decision Processes (BRMDPs). Theoretically, we demonstrate that RLBR can be addressed by solving a standard MDP with properly redistributed bagged rewards allocated to each instance within a bag. Empirically, we find that reward redistribution becomes more challenging as the bag length increases, due to reduced informational granularity. Existing reward redistribution methods are insufficient to address these challenges. Therefore, we propose a novel reward redistribution method equipped with a bidirectional attention mechanism, enabling the accurate interpretation of contextual nuances and temporal dependencies within each bag. We experimentally demonstrate that the proposed method consistently outperforms existing approaches. Codes are available in `https://github.com/Tang-Yuting/RLBR`.

---

[*]Equal contribution.

# 1 Introduction

Reinforcement Learning (RL) has achieved remarkable success in various domains, including autonomous driving (Kiran et al., 2021), healthcare (Yu et al., 2021), complex game playing (Silver et al., 2016; Wurman et al., 2022), and financial trading (Yang et al., 2020). One common and essential assumption for most RL algorithms is the availability of immediate reward feedback at each time step of the decision-making process. However, designing such immediate reward feedback is quite difficult and may mislead the policy learning in many real-world applications (Kwon et al., 2023; Lee et al., 2023). Recognizing this gap, numerous studies (Watkins, 1989; Ke et al., 2018; Arjona-Medina et al., 2019; Gangwani et al., 2020; Ren et al., 2021; Zhang et al., 2023) have explored the concept of RL with Trajectory Feedback (RLTF), primarily focusing on trajectory feedback where rewards are allocated at the end of a sequence.

However, real-world applications often feature complex non-immediate reward structures that traditional RLTF cannot fully capture. For instance, in autonomous driving (see Fig.1), rewards are typically linked to completing specific tasks by sequences of actions rather than individual actions or just the final objective (Early et al., 2022; Gaon & Brafman, 2020). Similarly, in mobile health interventions (Gao et al., 2024), the reward is defined by sustained daily commitment to being active throughout the intervention period, with multiple activity suggestions each day influencing this outcome. In such scenarios, providing rewards for every action is impractical, while focusing only on the final goal overlooks important aspects of the decision-making process. Previous studies have explored learning desirable policies using immediate rewards(Sutton & Barto, 1998; Vinyals et al., 2019) or trajectory-based feedback (Arjona-Medina et al., 2019; Gangwani et al., 2020; Ren et al., 2021) are not ideal methods for such scenarios.

To address these challenges, we introduce *RL from Bagged Rewards* (RLBR), a framework that better aligns with real-world scenarios by considering the cumulative effect of a series of instances. In RLBR, sequences of instances are defined as *bags*, each associated with a *bagged reward*. This framework encompasses both the traditional RL setting, where each bag contains only a single instance, and the trajectory feedback setting, where a bag spans the entire trajectory, as special cases. Furthermore, RLBR offers the potential to reduce the labeling workload by lessening the frequency of reward annotations. However, this benefit is balanced by increased learning complexity due to the reduced granularity of information.

In the RLBR framework, our focus is on leveraging bagged reward information to discern the significance of each instance within a bag and to understand the relationships among different bags. The challenge lies in accurately interpreting the contextual nuances within individual bags, as the instances within a bag would be time-dependent on each other and their contributions to the bagged reward vary. Given the importance of contexts in RLBR, we turn to the bidirectional attention mechanism (Seo et al., 2016; Vaswani et al., 2017; Devlin et al., 2019), renowned for its effectiveness in contextual understanding, especially for time-dependent data. Specifically, we propose a Transformer-based reward model, leveraging the bidirectional attention mechanism to adeptly interpret contexts within a bag and allocate rewards to each instance accurately. This model can be utilized to enhance general RL algorithms, such as Soft Actor-Critic (SAC) (Haarnoja et al., 2018), for environments with bagged rewards.

In summary, our contributions are threefold:

1. We introduce the RLBR framework as a general problem setting and formalize it using an extension of traditional Markov Decision Processes (MDPs), which we define as *Bagged Reward MDPs* (BRMDPs). This flexible reward structure bridges instance-level and trajectory-level rewards by supporting variable-length reward bags, making it suitable for real-world scenarios (Section 3).

2. We establish a theoretical connection between MDPs after reward redistribution and the original BRMDPs, providing a solid foundation for policy learning under bagged rewards. Building on this, we propose a Transformer-based reward model with a bidirectional attention mechanism that effectively captures temporal and contextual dependencies, combined with an alternating optimization algorithm to jointly enhance reward redistribution and policy learning (Section 4).

3. Extensive experiments show that our method not only outperforms existing approaches but also maintains robust performance as bag length increases, highlighting its ability to mimic ground truth reward distributions while exhibiting contextual understanding and adaptability to environmental dynamics (Section 5).

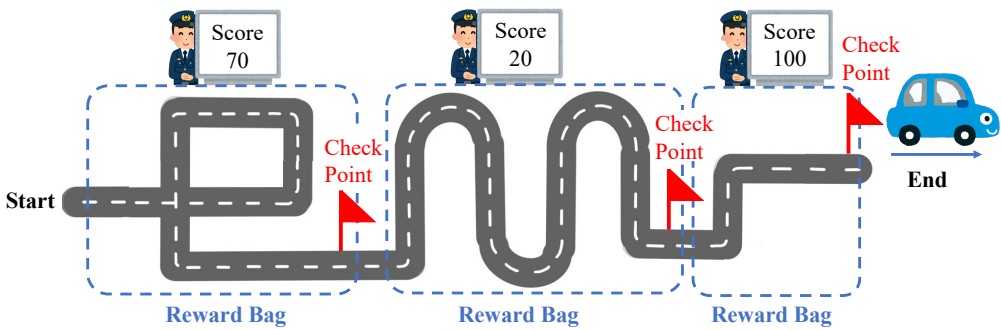

Figure 1: An illustration of the reward bag structure in an autonomous driving scenario. The driving trajectory is divided into multiple segments, or bags, each representing a sequence of actions that is evaluated and assigned a cumulative reward by an external evaluator.

## 2 Related Works

With the growing attention on RLTF, where trajectory feedback is often considered the sum of rewards over the entire trajectory, new methods specifically designed for RLTF have emerged. Return Decomposition for Delayed Rewards (RUDDER) (Arjona-Medina et al., 2019) used a return-equivalent formulation for precise credit assignment, with advancements incorporating expert demonstrations and language models (Liu et al., 2019; Widrich et al., 2021; Patil et al., 2022). Both Iterative Relative Credit Refinement (IRCR) (Gangwani et al., 2020) and Randomized Return Decomposition (RRD) (Ren et al., 2021) assumed that each state-action pair contributes equally to the trajectory feedback. Specifically, IRCR presented a uniform reward redistribution model, while RRD proposed a novel upper bound for return-equivalent assumptions, integrating return decomposition with uniform reward redistribution. Additionally, Han et al. (2022) proposed modifying RL algorithms to incorporate sequence-level information, enabling agents to learn from broader structures and long-term outcomes. On the other hand, some studies on sparse rewards, such as reward shaping (Ng et al., 1999; Tambwekar et al., 2019; Hu et al., 2020) and intrinsic reward methods (Sorg et al., 2010; Pathak et al., 2017; Zheng et al., 2018; 2021), could be directly applied to RLTF. However, these approaches primarily focused on addressing the exploration-exploitation trade-off in sparse reward settings, which made them less ideal for solving RLTF problems. A detailed discussion on the sparse reward settings and methods is provided in Appendix B.

Despite recent advances, methods that assume equal contribution from each state-action pair to trajectory feedback overlook the varying importance of individual instances. On the other hand, methods that consider sequence information struggled to capture effective information from long trajectories, resulting in suboptimal training outcomes, as evidenced by prior studies (Ren et al., 2021; Zhang et al., 2023). Building on previous methods, we adopt a reward redistribution learning strategy to enhance policy learning in the context of reward bags. The proposed method aims to extract information from bag-level rewards, taking into account the contribution of each instance, ultimately learning a reliable and well-performing policy. Experimental results (see Section 5) demonstrate that the proposed method can accurately capture information from long sequences, and the rich information in the bags further enhances its effectiveness.

## 3 Reinforcement Learning from Bagged Reward

In this section, we first provide preliminaries on RL with immediate rewards and trajectory feedback, and then formulate the RLBR problem with an extension of the traditional MDPs, the BRMDPs.

### 3.1 Preliminaries

In traditional RL problems, which are modeled using MDPs, each state-action pair is promptly associated with a reward (Sutton & Barto, 1998). This paradigm is encapsulated in a tuple $\mathcal{M} = (\mathcal{S}, \mathcal{A}, P, r, \mu)$, with

$\mathcal{S} \ni s$ and $\mathcal{A} \ni a$ as finite sets of states and actions, $P = p(s'|s, a)$ as a state transition probability function to state $s'$ when action $a$ is taken at state $s$, $r(s, a)$ as the immediate reward function when action $a$ is taken at state $s$, and $\mu = p(s)$ as the initial state distribution. The primary objective in this framework is to discover a policy $\pi = p(a|s)$, which is a conditional probability of taking action $a$ given state $s$, that maximizes the cumulative sum of rewards over a horizon length $T$:

$$J(\pi) = \mathbb{E}_{\pi, P, \mu}\left[\sum_{t=0}^{T-1} r(s_t, a_t)\right]. \tag{1}$$

Given that associating each state-action pair with a reward is challenging in real-world scenarios and involves significant labeling costs, RLTF, also known as episodic or delayed rewards in some works (Watkins, 1989; Arjona-Medina et al., 2019; Zhang et al., 2023), has become increasingly prominent in many applications (Agarwal et al., 2021; Chen et al., 2024). Distinct from traditional RL, RLTF offers only one feedback after a complete trajectory (Arjona-Medina et al., 2019; Zhang et al., 2023). A trajectory $\tau = \{(s_0, a_0), (s_1, a_1), \ldots, (s_{T-1}, a_{T-1})\}$ consists of $T$ state-action pairs, with a cumulative reward $R_{\text{traj}}(\tau)$ that is the sum of latent immediate rewards $\sum_{t=0}^{T-1} r(s_t, a_t)$, observable only at the end. Denoting by $\mathcal{T}(\pi)$ as the distribution of trajectories induced by $\pi, P, \mu$, the learning objective in RLTF is to maximize the expected trajectory-based reward:

$$J_{\text{traj}}(\pi) = \mathbb{E}_{\mathcal{T}(\pi)}\left[R_{\text{traj}}(\tau)\right]. \tag{2}$$

### 3.2 Problem Formulation

We formulate the RLBR problem as an extension of the traditional MDP, referred to as the BRMDP, which features a reward granularity that lies between the two aforementioned settings. We begin by defining the concept of *bags*, which are sub-pieces of complete trajectories. A trajectory $\tau$ is divided into several neighboring bags, and a bag of size $n_i$, which starts from time $i$, is defined as $B_{i,n_i} = \{(s_i, a_i), \ldots, (s_{i+n_i-1}, a_{i+n_i-1})\}, 0 \leq i \leq i + n_i - 1 \leq T - 1$. Afterward, we define the BRMDP to navigate the complexities of the aggregated non-Markovian reward signals:

**Definition 1** (BRMDP). *A BRMDP is defined by the tuple $(\mathcal{S}, \mathcal{A}, P, R, \mu)$, where*

- *$\mathcal{S}$ and $\mathcal{A}$ are finite sets of states and actions.*

- *$P$ is the state transition probability function.*

- *$R$ denotes the bagged reward function, which defines the reward over a bag: $R(B_{i,n_i})$.*

- *$\mu$ represents the initial state distribution.*

In the RLBR framework, a bag $B_{i,n_i}$ metaphorically a unified reward unit for a contiguous sequence of state-action pairs. Importantly, we do not impose any specific structural assumption on the bagged reward itself. It can be formed through various aggregation mechanisms beyond simple summation of instance-level reward, depending on the task and environment. A trajectory $\tau$ is a composite of a set of bags, denoted as $\mathcal{B}_\tau$, which ensures that each trajectory includes at least one reward bag. We further assume a bag partition function defined by the environment: $\mathcal{G} : \tau \mapsto \mathcal{B}_\tau$, which is a task-dependent function for generating bags given an input trajectory. Accordingly, the objective is to learn a policy $\pi$ that maximizes the expected cumulative bag-level reward:

$$J_{\text{B}}(\pi) = \mathbb{E}_{\mathcal{T}(\pi)}\left[\sum_{B \in \mathcal{B}_\tau} R(B)\Big|\mathcal{G}\right]. \tag{3}$$

Notably, the RLBR framework subsumes several existing paradigms as special cases. When each bag consists a single instance ($n_i = 1$ for $i = 0, 1, \ldots, T - 1$), RLBR simplifies to traditional RL. Conversely, when a single bag covers the entire trajectory ($n_0 = T$), RLBR reduces to RLTF. This adaptability highlights the flexibility of the RLBR framework in accommodating various reward structure scenarios.

# 4 Reward Redistribution for RLBR

This section delves into the proposed reward redistribution method for RLBR. We aim for the reward redistribution to meet the following criteria: first, the policies learned from the MDP after reward redistribution should be consistent with those learned from the BRMDP; second, the redistribution process should reflect each instance's contribution based on the sequence information. The subsequent subsections are structured to first theoretically prove the rationality of reward redistribution, followed by proposing the use of a bidirectional attention mechanism for this redistribution. Finally, we outline a comprehensive algorithm for developing efficient policies within the BRMDP framework.

## 4.1 Equivalence of Optimal Policies in the BRMDP and the MDP after Reward Redistribution

In the BRMDP, agents do not have direct access to the immediate rewards associated with each instance. To address this, we adopt a common approach used in trajectory feedback scenarios, modeling the cumulative sum form as the reward redistribution strategy to convert bagged rewards into instance-level rewards (Ren et al., 2021; Arjona-Medina et al., 2019; Zhang et al., 2023). Specifically, we define $R(B_{i,n_i}) = \sum_{t=i}^{i+n_i-1} \hat{r}(s_t, a_t)$, where $\hat{r}(s_t, a_t)$ represents the redistributed reward. However, a critical question arises: How can we ensure that the policy learned with redistributed rewards performs well in the original BRMDP? To address this, we provide the following theorem:

**Theorem 1.** *Consider a BRMDP where only the sequence-level bagged reward is available. Suppose the bagged reward is redistributed across instances while preserving its total sum within each bag. Under this assumption, the set of optimal policies $\Pi$ learned in the resulting MDP is identical to the set of optimal policies $\Pi_{\mathrm{B}}$ in the BRMDP, meaning that $\Pi = \Pi_{\mathrm{B}}$.*

*Proof.* Over a complete trajectory $\tau$, the cumulative reward in BRMDP can be expressed as the sum of the rewards from all the bags along the trajectory:

$$\sum_{B \in \mathcal{B}_\tau} R(B) = \sum_{i \in \mathcal{I}_\tau} R(B_{i,n_i}) = \sum_{i \in \mathcal{I}_\tau} \left( \sum_{t=i}^{i+n_i-1} \hat{r}(s_t, a_t) \right) = \sum_{t=0}^{T-1} \hat{r}(s_t, a_t),$$

where $\mathcal{I}_\tau$ denotes the set of initial timestep indices for $B \in \mathcal{B}_\tau$.

The policy optimization objective in BRMDP, $J_{\mathrm{B}}(\pi)$, aims to maximize the expected sum of bagged rewards along the trajectory from $t = 0$, which is equivalent to maximizing the cumulative reward in redistributed reward MDP, $J(\pi)$:

$$J_{\mathrm{B}}(\pi) = \mathbb{E}_{\mathcal{T}(\pi)} \left[ \sum_{B \in \mathcal{B}_\tau} R(B) \middle| \mathcal{G} \right] = \mathbb{E}_{\pi, P, \mu} \left[ \sum_{t=0}^{T-1} \hat{r}(s_t, a_t) \right] = J(\pi).$$

Given that the expected cumulative rewards for any policy in the original BRMDP and redistributed reward MDP frameworks are equivalent, under the condition of infinite exploration or exhaustive sampling within the state-action space, the sets of optimal policies for each framework also coincide, implying that $\Pi = \Pi_{\mathrm{B}}$.

$\square$

Under this theorem, directly using bagged rewards or applying an average reward redistribution strategy can, in theory, lead to an optimal policy. However, directly using bagged rewards introduces delayed feedback, which slows down policy convergence, as demonstrated in Arjona-Medina et al. (2019). Meanwhile, average reward redistribution assumes equal contributions from all instances within a bag, potentially diluting critical decision points and consequently affecting policy learning (Gangwani et al., 2020; Ren et al., 2021). Therefore, an effective reward redistribution method should capture the actual contributions of individual state-action pairs within each bag. By assigning rewards in a way that reflects their true impact on bagged rewards, we can enhance credit assignment accuracy and improve the quality of policy optimization. By this theorem, we ensure that optimizing a policy on an MDP after rewards redistribution is equivalent to optimizing on the original BRMDP. Below, we introduce our specific reward redistribution approach.

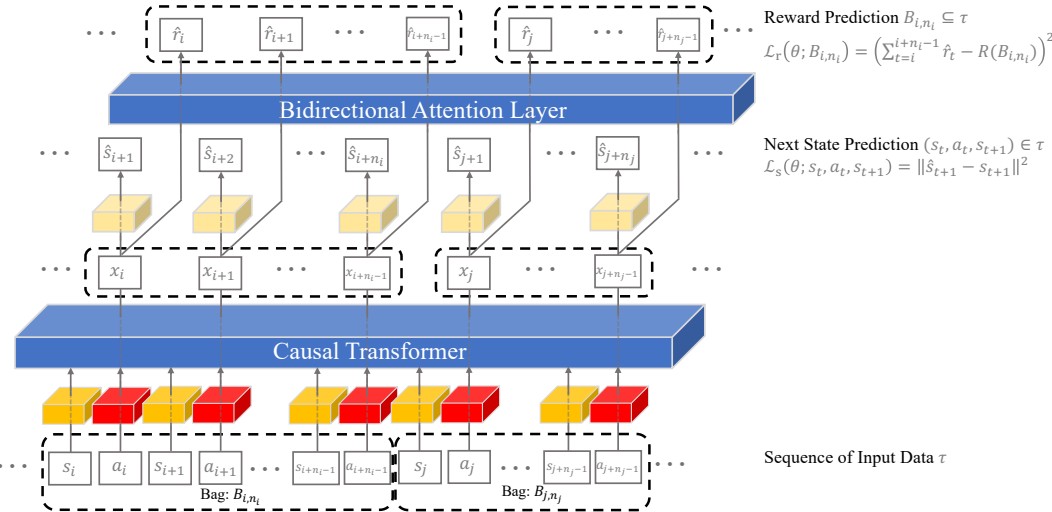

Figure 2: The illustration of the Reward Bag Transformer (RBT) architecture. The Causal Transformer is used for reward representation by processing sequences of input data consisting of state-action pairs. The bidirectional attention layer is used for reward redistribution, utilizing the outputs of the Causal Transformer to predict instance-level rewards. The next state prediction helps the model understand the environment, thereby improving reward prediction.

## 4.2 Reward Redistribution based on Bidirectional Attention Mechanism

Building on Theorem 1, which establishes that optimal policies in the BRMDP and the MDP after reward redistribution are equivalent, our focus shifts to the crucial process of redistributing rewards within a bag. To capture the contextual influence of each instance within the sequence, the Causal Transformer (Vaswani et al., 2017) naturally serves as a suitable sequential prediction model. Traditionally, Transformers in RL have been used in a unidirectional manner (Chen et al., 2021; Janner et al., 2021; Micheli et al., 2023), where only previous instances influence the current prediction due to the unobservability of future instances. However, since our bagged rewards are generally non-Markovian, both preceding and subsequent instances can influence the contribution of the current instance to the bagged reward. Therefore, understanding the relationships among instances within a bag is therefore pivotal. This insight leads us to employ a *bidirectional* attention mechanism (Seo et al., 2016; Vaswani et al., 2017; Devlin et al., 2019) to capture these relationships. This mechanism connects both past and future instances within a bag, enabling a more comprehensive understanding of contextual influences. By quantitatively evaluating the contribution of each instance, the bidirectional attention mechanism facilitates nuanced reward redistribution, as demonstrated experimentally in Section 5.

## 4.3 Reward Bag Transformer

We introduce the *Reward Bag Transformer* (RBT), a novel reward redistribution method for the BRMDP. The RBT is engineered to comprehend the complex dynamics of the environment through bags and to precisely predict instance-level rewards, facilitating effective reward redistribution.

### 4.3.1 Causal Transformer for Reward Representation

Referring to Fig. 2, the RBT comprises a Causal Transformer (Vaswani et al., 2017; Radford et al., 2018), which maintains the chronological order of state-action pairs (Chen et al., 2021; Janner et al., 2021). For each time step $t$ in a sequence of $M$ time steps, the Causal Transformer, represented as a function $f$, processes the input sequence $\sigma = \{s_0, a_0, \ldots, s_{M-1}, a_{M-1}\}$, generating the output $\{x_t\}_{t=0}^{M-1} = f(\sigma)$. By aligning the output head $x_t$ with the action token $a_t$, we directly model the consequence of actions, which

---

**Algorithm 1** Policy Optimization with RBT

---

1: Initialize replay buffer $\mathcal{D}$, RBT parameters $\theta$.
2: **for** trajectory $\tau$ collected from the environment **do**
3:    Store trajectory $\tau$ with bag information $\{(B_{i,n_i}, R(B_{i,n_i}))\}_{B_{i,n_i} \in \mathcal{B}_\tau}$ in $\mathcal{D}$.
4:    Sample batches from $\mathcal{D}$.
5:    Estimate bag loss based on Eq. equation 7.
6:    Update RBT parameters $\theta$ based on the loss.
7:    Relabel rewards in $\mathcal{D}$ using the updated RBT.
8:    Optimize policy using the relabeled data by off-the-shelf RL algorithms (such as SAC (Haarnoja et al., 2018)).
9: **end for**

---

are pivotal in computing immediate rewards and predicting subsequent states, thereby helping the model better understand environmental dynamics.

### 4.3.2 Bidirectional Attention Layer for Reward Redistribution

Once we have obtained the output embeddings $\{x_t\}_{t=0}^{M-1}$, for reward prediction, they pass through a bidirectional attention layer to produce $\{\hat{r}_t\}_{t=0}^{M-1}$, where $\hat{r}_t \equiv \hat{r}_\theta(s_t, a_t)$ with $\theta$ being the RBT parameters. This layer addresses the unidirectional limitation of the Causal Transformer architecture (Vaswani et al., 2017; Radford et al., 2018), integrating past and future contexts for enhanced reward prediction accuracy. For state prediction, since the state at step $t + 1$ depends only on the previous steps, $x_t$ is input into a state linear decoder, yielding the predicted next state $\hat{s}_{t+1} \equiv \hat{s}_\theta(s_t, a_t)$.

The core of the RBT architecture is its bidirectional attention mechanism. For each output embedding $x_t$, we apply three different linear transformations to obtain the query embedding $\mathbf{q}_t \in \mathbb{R}^d$, key embedding $\mathbf{k}_t \in \mathbb{R}^d$, and value embedding $v_t \in \mathbb{R}$, where $d$ is the embedding dimension of the key. The instance-level reward is then calculated by

$$\hat{r}_t = \sum_{\ell=0}^{M-1} \texttt{softmax}(\frac{\{\langle \mathbf{q}_t, \mathbf{k}_{t'} \rangle\}_{t'=0}^{M-1}}{\sqrt{d}})_\ell \cdot v_\ell. \tag{4}$$

The rescaling operation is used to prevent extremely small gradients as in Vaswani et al. (2017). This mechanism enables the RBT to consider both the immediate and contextual relevance of each instance in a bag when predicting rewards.

### 4.4 Learning Objectives

The learning objectives of the RBT are twofold: (1) *reward prediction* within each reward bag and (2) *state transition forecasting*. These objectives are critical for enabling the model to navigate the complex dynamics of BRMDP environments.

### 4.4.1 Reward Prediction

The RBT is trained to ensure that, for each reward bag, the sum of predicted instance-level rewards matches the total bagged reward. This is vital for maintaining the integrity of the reward structure in the BRMDP framework. The loss function is expressed as

$$\mathcal{L}_{\mathrm{r}}(\theta; B_{i,n_i}) = \left( \sum_{t=i}^{i+n_i-1} \hat{r}_t - R(B_{i,n_i}) \right)^2. \tag{5}$$

This loss function encourages RBT to learn a meaningful distribution of rewards across instances within a bag while ensuring that the total redistributed reward remains consistent with the original bagged reward. By enforcing this consistency, the model aligns with the conditions established in Theorem 1.

### 4.4.2 State Transition Forecasting

Alongside reward prediction, the RBT is tasked with accurately predicting the next state in the environment given the current state and action. This capability is crucial for understanding the dynamics of the environment, which in turn enhances the accuracy of reward prediction. The corresponding loss function is

$$\mathcal{L}_{\text{s}}(\theta; s_t, a_t, s_{t+1}) = \|\hat{s}_{t+1} - s_{t+1}\|^2, \tag{6}$$

where $\| \cdot \|$ denotes the $\ell^2$-norm. This loss emphasizes the model's understanding of dynamics.

### 4.4.3 Composite Loss

The final learning objective combines the reward and state prediction losses:

$$\begin{aligned}
\mathcal{L}_{\text{bag}}(\theta) = &\underset{\tau \sim \mathcal{D}}{\mathbb{E}}\big[\mathcal{L}_{\text{r}}(\theta; B_{i,n_i})\big|B_{i,n_i} \in \mathcal{B}_\tau\big] \\
&+ \beta \underset{\tau \sim \mathcal{D}}{\mathbb{E}}\big[\mathcal{L}_{\text{s}}(\theta; s_t, a_t, s_{t+1})\big|(s_t, a_t, s_{t+1}) \in \tau\big],
\end{aligned} \tag{7}$$

where $\beta > 0$ is a balancing coefficient between the two loss components, and $\mathcal{D}$ denotes the replay buffer.

The RBT's dual predictive capacity is its key advantage, enabling precise reward redistribution to individual instances and forecasting the next state. This leverages environmental dynamics for enhanced reward distribution as experimentally evidenced in Section 5. Integrated with off-the-shelf RL algorithms such as SAC (Haarnoja et al., 2018), the RBT can enhance policy learning within the BRMDP framework, as outlined in Algorithm 1.

## 5 Experiment

In the following experiment section, we scrutinize the efficacy of our proposed method using benchmark tasks from both the MuJoCo (Brockman et al., 2016) and the DeepMind Control Suite (Tassa et al., 2018) environments, focusing on scenarios with bagged rewards. Initially, we assess the performance of our method to understand its overall effectiveness. Subsequently, we examine whether the proposed RBT reward model accurately predicts rewards. Finally, we evaluate the indispensability of each component of the reward model, questioning if every part is essential.

### 5.1 Performance Comparison

To evaluate the performance of our method, we conduct a detailed comparison across multiple baselines. This analysis helps us understand how our approach fares against existing methods in handling scenarios with bagged rewards.

### 5.1.1 Experiment Setting

We evaluated our method on benchmark tasks from the MuJoCo locomotion suite (Ant-v2, Hopper-v2, HalfCheetah-v2, and Walker2d-v2) and the DeepMind Control Suite (cheetah-run, quadruped-walk, fish-upright, cartpole-swingup, ball_in_cup-catch, and reacher-hard).

In standard reinforcement learning settings, rewards are assigned at each step based on immediate state-action transitions. However, in our setup, only the bagged reward is observable, while the individual rewards for each instance remain unknown. Bagged rewards are constructed by aggregating the rewards of all state-action pairs within a predefined sequence. Specifically, in the main text, we present experimental results using summation-based aggregation, while additional results for more complex reward structures are provided in Appendix C. To implement this reward structure, the bagged reward is assigned to the last instance within the bag, while all other state-action pairs within the bag receive a reward of zero. This setting effectively simulates delayed or sparse reward scenarios, making the learning problem more challenging due to the lack of direct instance-level reward supervision. For each task, we collected 1e6 time steps, while the maximum episode length was fixed at 1,000 steps, ensuring consistency across experiments.

### 5.1.2    Baselines

In the comparative analysis, our framework was rigorously evaluated against several leading algorithms in the domain of RLTF:

- **SAC** (Haarnoja et al., 2018): It directly utilized the original bagged reward information for policy training using the SAC algorithm.

- **IRCR** (Gangwani et al., 2020): It adopted a non-parametric uniform reward redistribution approach. We have adapted IRCR for bagged reward setting.

- **RRD** (Ren et al., 2021): It employed a reward model trained with a randomized return decomposition loss. We have adapted RRD for bagged reward setting.

- **HC** (Han et al., 2022): The HC-decomposition framework was utilized to train the policy using a value function that operates on sequences of data. We employed the code as provided by the original paper.

- **Shaping** (Hu et al., 2020): A widely used reward shaping method for sparse reward setting, the shaping reward function is adopted from Hu et al. (2020).

- **LIRPG** (Zheng et al., 2018): It learned an intrinsic reward function to complement sparse environmental feedback, training policies to maximize combined extrinsic and intrinsic rewards. We use the same code provided by the paper.

While methods like RUDDER (Arjona-Medina et al., 2019) and Align-RUDDER (Patil et al., 2022) are known for addressing the problem of trajectory feedback, previous studies (Gangwani et al., 2020; Ren et al., 2021; Zhang et al., 2023) have shown superior performance using referenced methods. Additionally, since Align-RUDDER relies on successful trajectories for scoring state-action pairs, which is impractical in MuJoCo (Patil et al., 2022), we ultimately excluded both methods from our comparison. Sparse reward baselines were included in our comparisons to highlight the distinctions and challenges unique to our bagged reward setting. While both settings provide rewards after a sequence, sparse rewards are Markovian, whereas our bagged rewards are non-Markovian. Therefore, including sparse reward baselines illustrates that directly applying sparse reward methods to bagged rewards is possible but not effective. For more discussion on sparse rewards, see Appendix B. Besides, detailed descriptions of the model parameters and hyper-parameters used during training are provided in Appendix A, with more experimental results available in Appendix C.

### 5.1.3    Evaluation Metric

We report the average accumulative reward across 6 seeds with random initialization to demonstrate the performance of evaluated methods. Higher accumulative reward in evaluation indicates better performance.

### 5.1.4    Experimental Results

In the **fixed-length** reward bag experiment (see Fig. 3), we conducted experiments with six bag lengths (5, 25, 50, 100, 200, and 500) and trajectory feedback (labeled as 9999) across each environment, where bagged reward is made by accumulating immediate rewards within bags. This aimed to illustrate the influence of varying bag lengths, providing insight into how bag size affected the performance of the learning algorithm. The SAC method, using bagged rewards directly from the environment, suffers from a lack of guidance in agent training due to missing immediate reward information. This issue worsens with longer bag lengths, indicating that increased reward sparsity leads to less effective policy optimization. The IRCR and RRD methods, treating rewards uniformly within a reward bag, outperform SAC, suggesting benefits from even approximate guidance. However, notable variance in their results indicates potential consistency and reliability issues. The HC method excels only with shorter bag lengths, suggesting that this value function modification method struggles to utilize information from longer sequences. The Shaping and LIRPG methods exhibit subdued performance across tasks, as they are designed to solve sparse reward problems with

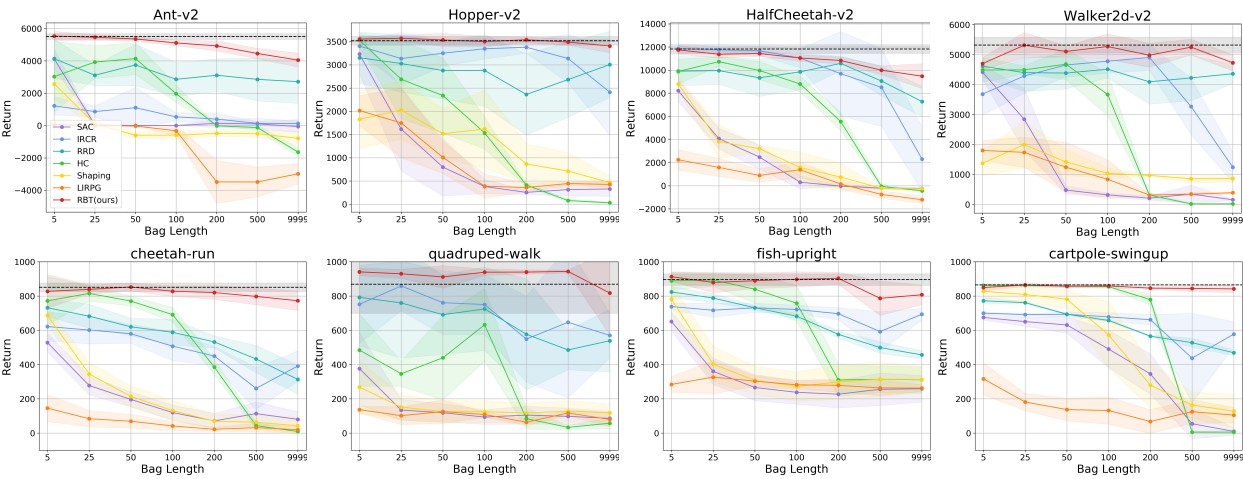

Figure 3: Performance comparison across fixed-length reward bag in MuJoCo (top row) and DeepMind Control Suite (bottom row) environments with six different length bag settings (5, 25, 50, 100, 200, and 500) and trajectory feedback (labeled as 9999). The mean and standard deviation are computed over 6 trials with different random seeds across a total of 1e6 time steps. Additionally, the black dashed line in the figures represents the performance of vanilla SAC with a bag length of 1, along with its standard deviation, serving as a reference baseline.

Markovian rewards, which do not align with the reward bag setting. The proposed RBT method consistently outperforms the other approaches across all the environments and bag lengths, showing that it is not only well-suited for environments with short reward bags but also capable of handling large reward bag scenarios. This demonstrated the capability of RBT to learn from the sequence of instances and, by integrating bagged reward information, accurately allocate rewards to instances, thereby guiding better policy training. Notably, for shorter bag lengths, the performance of the RBT method closely matches that of vanilla SAC. This suggests that RBT effectively preserves critical reward signals, allowing the agent to achieve performance comparable to the vanilla SAC baseline. In addition to experiments conducted under settings consistent with previous work (Gangwani et al., 2020; Ren et al., 2021; Zhang et al., 2023), where the bagged reward is the sum of immediate rewards, we also tested bagged rewards with more complex structures (see Appendix C.2). These results further demonstrated that our method outperforms the baselines, indicating that previous approaches are effective only when bagged rewards are the sum of immediate rewards, highlighting their limitations.

To validate the effectiveness of our approach under more complex conditions, we designed an experiment with **arbitrary-length** reward bags, allowing not only for variations in bag lengths but also for overlaps and gaps between reward bags. This setup simulated more realistic scenarios and tested the robustness of our method. The results, detailed in Table 1, confirm the superior performance of the proposed RBT method in these complex reward settings. These findings suggest RBT's potential for broader applications, demonstrating its versatility and robustness in handling intricate reward dynamics.

We also conducted experiments in **sparse reward environments**, where tasks are rewarded with 1 at the end of the trajectory if completed and 0 otherwise. In this setting, the reward does not accumulate as in our redistribution model. As shown in Table 2, RLTF methods perform poorly in sparse reward environments. In contrast, our proposed method not only outperforms RLTF baselines but also exceeds the performance of sparse reward baselines. From another perspective, we also tested the performance of sparse reward methods in the bagged reward setting (see Appendix B). The results indicated that, due to differences in the reward structure, sparse reward methods do not adapt well in the bagged reward setting either. This demonstrates that our proposed RBT model can effectively capture relationships between instances across different reward

Table 1: Performance comparison across arbitrary-length reward bag configurations over 6 trials with 1e6 time steps for training, presenting average scores and standard deviations. "Narrow" refers to bags with lengths varying arbitrarily from 25 to 200 and narrow intervals between -10 to 10. "Broad" denotes the setting with bag lengths varying arbitrarily from 100 to 200 and broad interval variations from -40 to 40. The best and comparable methods based on the paired t-test at the significance level 5% are highlighted in boldface.

| Bag Setting | Environment | SAC | IRCR | RRD | HC | Shaping | LIRPG | RBT(ours) |
|---|---|---|---|---|---|---|---|---|
| Narrow | Ant-v2 | 0.87 (2.98) | 368.69 (119.74) | 2272.39 (835.86) | 106.92 (153.86) | -732.03 (225.75) | -756.78 (763.66) | **5122.50 (206.44)** |
| | Hopper-v2 | 317.72 (52.17) | 3353.35 (61.97) | 2184.41 (807.71) | 510.66 (94.49) | 1031.38 (335.05) | 126.13 (30.18) | **3499.54 (76.62)** |
| | HalfCheetah-v2 | 788.45 (1737.57) | **10853.85 (573.72)** | 9709.62 (1479.73) | 4027.25 (441.01) | 2348.78 (390.48) | 1101.38 (1248.45) | **11282.24 (266.08)** |
| | Walker2d-v2 | 193.07 (48.40) | 4144.65 (673.66) | 3536.90 (546.66) | 309.19 (171.69) | 1022.32 (369.81) | 123.43 (50.97) | **4983.39 (311.09)** |
| Broad | Ant-v2 | -3.31 (4.15) | 368.69 (158.46) | 1323.50 (1079.60) | 5.97 (20.08) | -634.69 (161.57) | -1264.08 (416.86) | **5167.79 (303.83)** |
| | Hopper-v2 | 329.48 (44.21) | 3296.20 (216.35) | 1102.38 (892.12) | 701.84 (149.44) | 1066.51 (429.68) | 203.01 (177.80) | **3499.53 (94.00)** |
| | HalfCheetah-v2 | 43.96 (94.32) | 9158.14 (1402.62) | 4199.16 (1476.85) | 4460.80 (518.94) | 1459.83 (1251.52) | 924.26 (1110.97) | **10837.15 (254.99)** |
| | Walker2d-v2 | 176.09 (49.81) | 4179.08 (937.42) | 330.96 (79.26) | 447.45 (155.63) | 947.87 (90.46) | 194.95 (98.05) | **5202.38 (248.35)** |

Table 2: Performance comparison on sparse reward environments over 6 trials with 1e6 time steps for training, presenting average scores and standard deviations. Rewards are given with 1 at the end of the trajectory if completed and 0 otherwise. The best and comparable methods based on the paired t-test at the significance level 5% are highlighted in boldface.

| Environment | SAC | IRCR | RRD | HC | Shaping | LIRPG | RBT(ours) |
|---|---|---|---|---|---|---|---|
| ball_in_cup-catch | 646.90 (231.15) | 773.19 (73.06) | 666.18 (33.02) | 76.19 (67.25) | 847.88 (178.01) | 125.50 (21.25) | **972.19 (5.50)** |
| reacher-hard | 531.88 (127.94) | 584.76 (269.32) | 347.16 (106.56) | 7.57 (3.50) | 664.49 (170.50) | 8.74 (2.36) | **735.67 (144.90)** |

structures. By accurately distributing reward signals across instances, it provides more informative feedback, enabling the agent to learn more efficiently and ultimately improving overall policy performance.

## 5.2 Case Study

The previous experimental results showcase the superiority of RBT over baselines. This led to an intriguing inquiry: Is the RBT reward model proficient in accurately redistributing rewards? To investigate this question, we performed an experiment focused on reward comparison, utilizing a trajectory generated by an agent trained in the Hopper-v2 environment with a bag length of 100, where the bagged reward represents an accumulated reward. As shown in Fig. 4, which spans 1000 steps, RBT-predicted rewards, unobservable true rewards, and observable bagged rewards (presented in a uniform format for better visualization) are compared. The figure indicates that the rewards predicted by the RBT closely match the trends of the true rewards. This observation suggest that the RBT is effective at reconstructing true rewards from bagged rewards, despite the coarse nature of the environmental reward signals.

Beneath the figure, a series of images depicts a complete jump cycle by the agent, illustrating its motion sequence: mid-air, landing, jumping, and returning to mid-air. Red boxes highlight specific states that correspond to reward peaks and troughs, representing moments of maximum, minimum, and moderate rewards. In the Hopper-v2 environment, rewards consist of a constant "healthy reward" for operational integrity, a "forward reward" for progress in the positive x-direction, and a "control cost" for penalizing large

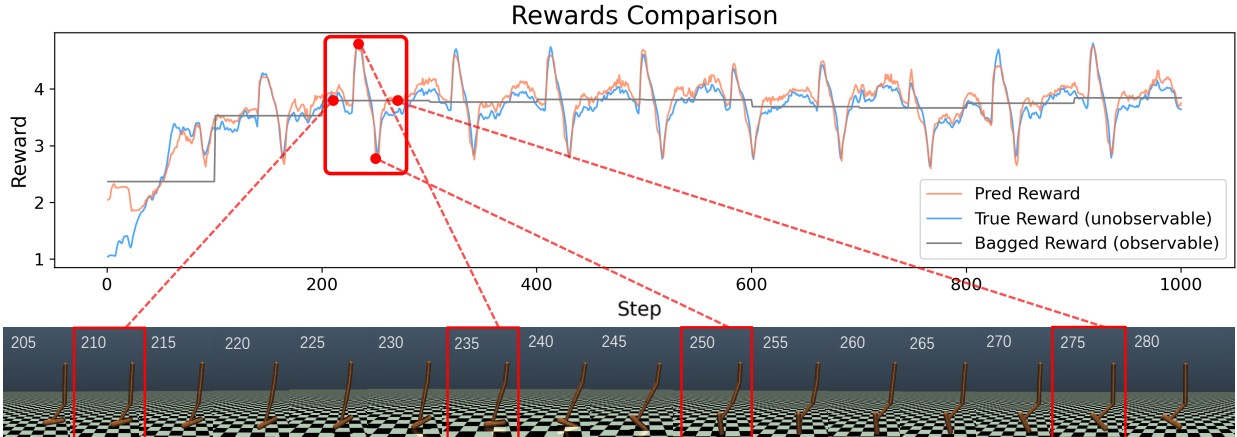

Figure 4: Rewards comparison and agent states in a trajectory with a bag length of 100 in the Hopper-v2 environment. The top graph compares predicted rewards against true rewards and aggregated bagged rewards. The bottom images show the Hopper agent's motion at different time steps, with the highlighted frames representing key moments that provide insight into the relationship between agent behavior and reward dynamics.

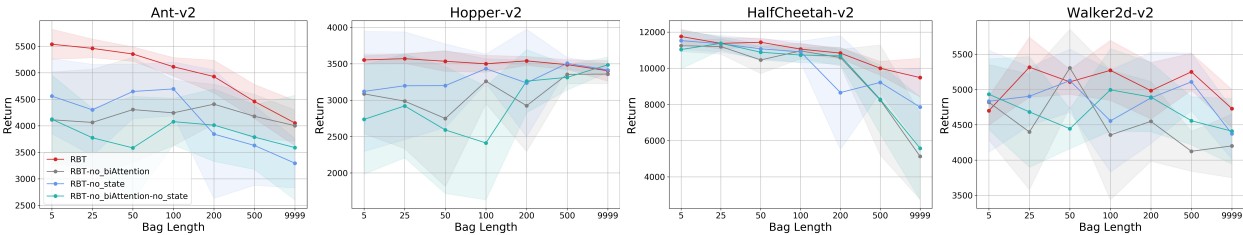

Figure 5: Ablation study of reward model components across various environments. The chart presents mean and standard deviation of rewards over 6 trials with 1e6 timesteps, showcasing the efficacy of the full proposed method relative to its variants without certain features.

actions. At peak reward instances, the agent is typically fully grounded in an optimal posture for forward leaping, which maximizes the "forward reward" through pronounced x-direction movement. Concurrently, it sustains the "healthy reward" and minimizes "control cost" through measured, efficient actions. This analysis underscores that the RBT can adeptly decode the environmental dynamics and the nuanced reward redistribution even under the setting of RLBR.

## 5.3 Ablation Study

We conducted comparisons to examine the role of RBT's modules. As shown in Fig. 5, the full RBT model consistently outperforms its variants, indicating a synergistic effect when all components are used together. Performance drops significantly when the bidirectional attention mechanism is removed, especially in complex environments like Ant-v2 and HalfCheetah-v2, suggesting its critical role in accurate reward prediction. Additionally, we can observe that removing the next state prediction component weakens the reward model's understanding of environmental dynamics, reducing reward prediction accuracy and hindering policy learning. The greatest performance decline occurs when both the next state prediction and bidirectional self-attention mechanism are absent, underscoring their individual and combined importance in building a robust reward model.

# 6 Discussion

In this section, we discuss the limitations of our approach, as well as its broader applicability and potential directions for future research.

## 6.1 Limitations

One major limitation of our approach is the training duration, which is longer compared to methods that do not require a reward model (see Appendix C.5). The increased computational cost primarily arises from the reward relabeling process, where the model updates assigned rewards in the replay buffer based on the learned reward distribution. This step adds significant overhead, making training less efficient, particularly for longer bag lengths, where reward prediction becomes more challenging. A potential direction for improving computational efficiency is prioritized experience replay (Schaul et al., 2015), which could selectively relabel more informative samples rather than processing the entire buffer equally, thereby reducing unnecessary computation. Another possible enhancement is a cached reward update mechanism, where only a subset of stored samples is updated per iteration instead of recomputing rewards for the full replay buffer. These optimizations have the potential to reduce training time while maintaining learning performance.

Another limitation of our approach is the assumption that the bag partition function $\mathcal{G}$ is provided by the environment. While this assumption simplifies the problem setting and allows us to focus on reward redistribution, it may not always hold in real-world scenarios. In cases where bag boundaries are unknown, an additional inference step would be required to identify bag transitions, potentially increasing the complexity of both learning and computational cost. Future work could explore methods for learning bag partitioning dynamically, such as incorporating segmentation models (Adams & MacKay, 2007; Truong et al., 2020), leveraging latent variable approaches (Yu, 2010; Kingma et al., 2013; Johnson & Willsky, 2013), or attention-based mechanisms (Yi et al., 2021) to infer bag structures from data.

Additionally, our current study does not explicitly address the generalization ability of the learned reward redistribution model across different environments. All experiments involve training and evaluating policies within individual environments without testing domain transfer capabilities. While this setup ensures a controlled evaluation, it remains unclear how well the model would adapt to new domains with different reward structures. Future work could investigate techniques such as meta-learning (Finn et al., 2017; Zintgraf et al., 2019) or domain adaptation (Higgins et al., 2017; Peng et al., 2018) to improve generalization.

Finally, our experiments primarily focus on the Control Suite. While this setting provides a clean testbed for evaluating our method, real-world problems often involve partial observability, sharp reward transitions, and challenging exploration scenarios. In environments where observations are incomplete or noisy, additional techniques such as belief state estimation (Kaelbling et al., 1998; Murphy, 2000; Igl et al., 2018) may be required to improve robustness. For environments with sparse and sharp reward transitions, a possible approach is to define bag boundaries based on reward occurrences, as done in our sparse reward experiments. Morever, adaptive bag partitioning (Adams & MacKay, 2007; Truong et al., 2020; Yu, 2010; Kingma et al., 2013; Johnson & Willsky, 2013) could be explored to better align reward redistribution with critical decision points. Lastly, our method focuses on credit assignment rather than exploration. In environments with difficult exploration, additional techniques such as intrinsic motivation (Schmidhuber, 1991; Pathak et al., 2017; Burda et al., 2019a) or entropy-based exploration (Hazan et al., 2019; Lee et al., 2021) could be integrated to improve performance.

## 6.2 Broader Applicability and Future Directions

A promising direction for extending our work is to apply the RLBR framework to practical scenarios involving both embodied and digital agents. Many real-world tasks in these domains can be naturally segmented into contiguous phases, each consisting of a sequence of interrelated actions that jointly contribute to intermediate progress toward the final goal (Kulkarni et al., 2016; Nachum et al., 2018). In embodied agents (Parr & Russell, 1997), such as household robots, tasks like cleaning, assembling, or navigation consist of successive physical interactions that can be grouped into temporally coherent segments based on spatial or behavioral continuity, making them well-suited for modeling with the RLBR framework. In digital agents (Peng et al.,

2017; Shi et al., 2019), such as dialogue systems or web-based task solvers, feedback is often delayed and context-dependent, typically emerging only after a sequence of actions achieves a sub-goal. For example, positive feedback may appear only after several dialogue turns. This makes per-action credit assignment challenging, whereas attributing rewards to contiguous segments better captures their joint contribution and improves learning stability. This structure aligns well with the bagged reward formulation, where reward signals are assigned to temporally localized segments rather than individual steps or the entire trajectory.

Given this structural alignment, we believe the RLBR framework, which focuses on an intermediate form of reward between instance-level and trajectory-level feedback, holds strong potential for real-world applications. It can serve as a bridge between low-level control and high-level task abstraction. Future work may explore integrating RLBR with hierarchical reinforcement learning (Barto & Mahadevan, 2003; Nachum et al., 2018) or jointly learning bag partitioning and policy optimization to support more adaptive behavior in complex domains.

## 7 Conclusion

In this paper, we introduced a novel learning framework called Reinforcement Learning from Bagged Rewards (RLBR). To address the challenges posed by this framework, we established theoretical connections between Bagged Reward MDPs (BRMDPs) and original MDPs after reward redistribution, providing a solid foundation for our approach. Based on this theory, we proposed a reward model, the Reward Bag Transformer (RBT), designed to redistribute rewards by interpreting contextual information within bags and understanding environmental dynamics. The efficacy of RBT was demonstrated through extensive experiments, where it consistently outperformed existing methods across various reward bag scenarios. Additionally, our case studies highlighted RBT's ability to effectively reallocate rewards while maintaining fidelity to the underlying true reward structure.

**Acknowledgments**

YT was supported by Institute for AI and Beyond, UTokyo. XC was supported by JSPS, KAKENHI Grant Number JP24KJ0610, Japan. YD was supported by National Natural Science Foundation of China, Grant Number 62206245. MS was supported by JST ASPIRE Grant Number JPMJAP2405, and Institute for AI and Beyond, UTokyo. The authors would like to thank anonymous reviewers for their insightful reviews.

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

## A    Experiment Settings and Implementation Details

**Benchmarks with Bagged Rewards.**    We introduced a novel problem setting in the suite of MuJoCo and DeepMind Control Suite locomotion benchmark tasks, termed as bagged rewards. Our simulations ran on the OpenAI Gym platform (Brockman et al., 2016) and the DeepMind Control Suite (Tassa et al., 2018), featuring tasks that stretched over long horizons with a set maximum trajectory length of $T = 1000$. We used MuJoCo version 2.0 for our simulations, which is available at `http://www.mujoco.org/`. MuJoCo is licensed under a commercial license, and we have adhered to its terms of service and licensing agreements as stated on the official website. The DeepMind Control Suite is available under an Apache License 2.0, and we have complied with its terms of use.

Reward bag experiments of different bag sizes (5, 25, 50, 100, 200, and 500) and trajectory feedback were set up to verify the effectiveness of the method. To evaluate the efficacy of proposed method, commonly used trajectory feedback algorithms were adapted to fit the bagged reward setting as baselines. In these experiments, each reward bag was treated as an individual trajectory, and these modified algorithms were applied accordingly. Additionally, experiments using standard trajectory feedback were conducted to provide a comparative baseline within the unique setting. The total episodic feedback was computed at the end of the trajectory and was the sum of the per-step rewards the agent had collected throughout the episode. This experiment setting was the same as some previous works for learning from trajectory feedback (Gangwani et al., 2020; Ren et al., 2021).

For the sparse reward experiments, the ball_in_cup-catch task provides a sparse reward of 1 when the ball is in the cup and 0 otherwise; in the reacher-hard task, the reward is 1 when the end effector penetrates the target sphere (Tassa et al., 2018). Therefore, we treated the sparse reward as trajectory feedback and conducted experiments where the entire trajectory was considered as a single reward bag in these environments. The bagged reward was identical to the sparse reward provided by the environment.

Table 3: Hyper-parameters of RBT.

| Hyper-parameter | Value |
|---|---|
| Number of Causal Transformer layers | 3 |
| Number of bidirectional attention layers | 1 |
| Number of attention heads | 4 |
| Embedding dimension | 256 |
| Batch size | 64 |
| Dropout rate | 0.1 |
| Learning rate | 0.0001 |
| Optimizer | AdamW (Loshchilov & Hutter, 2018) |
| Weight decay | 0.0001 |
| Warmup steps | 100 |
| Total gradient steps | 10000 |

**Implementation Details and Hyper-parameter Configuration.**    In our experiments, the policy optimization module was implemented based on soft actor-critic (SAC) (Haarnoja et al., 2018). We evaluated the performance of our proposed methods with the same configuration of hyper-parameters in all environments. The back-end SAC followed the JaxRL implementation (Kostrikov, 2021), which is available under the MIT License.

The RBT reward model was developed based on the GPT implementation in JAX (Frostig et al., 2018), which is available under the Apache License 2.0. Our experiments utilized the Causal Transformer with three layers and four self-attention heads, followed by a bidirectional self-attention layer with one self-attention head. For detailed hyper-parameter settings of the RBT, please refer to Table 3.

For the baseline methods, the IRCR (Gangwani et al., 2020) method was implemented based on the descriptions provided in the original paper. The RRD (Ren et al., 2021) and LIRPG (Zheng et al., 2018)

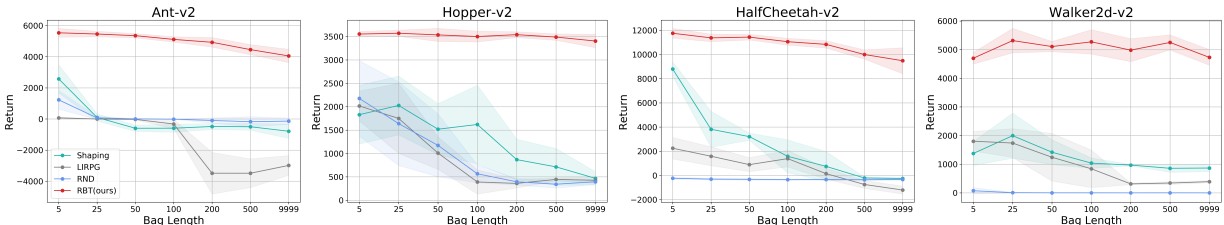

Figure 6: Comparison of proposed method and sparse reward baselines. The mean and standard deviation are displayed over 6 trials with different random seeds across a total of 1e6 time steps.

methods are both licensed under the MIT License. The code of HC (Han et al., 2022) is available in the supplementary material at `https://openreview.net/forum?id=nsjkNB2oKsQ`.

To ensure uniformity in the policy optimization process across all methodologies, each was subjected to 1,000,000 training iterations. For the proposed method, we initially collated a dataset comprising 10,000 time steps to pre-train the reward model. This model then underwent 100 pre-training iterations, a step deemed essential to adequately prepare the reward model before embarking on the principal policy learning phase. Following this initial warm-up period, the reward model was trained for 10 iterations after each new trajectory was incorporated. Moreover, to systematically gauge performance and progress, evaluations were carried out at intervals of every 5,000 time steps. The computational resources for these procedures were NVIDIA GeForce RTX 2080 Ti GPU clusters with 8GB of memory, dedicated to training and evaluating tasks.

## B  Discussion on Sparse Reward

The sparse reward setting presents significant challenges due to infrequent feedback, making it difficult for agents to effectively explore the environment and discover successful strategies. To address this challenge, various methods have been developed to enhance exploration. Reward shaping strategies (Ng et al., 1999; Hu et al., 2020; Tambwekar et al., 2019) added rewards to actions in a way that guides the agent towards better policies without altering the original reward function. Curiosity-driven methods (Pathak et al., 2017; Sekar et al., 2020) encouraged agents to explore the environment by visiting unseen states, potentially solving tasks with sparse rewards. Additionally, curriculum learning in RL (Florensa et al., 2018; Riedmiller et al., 2018) involved presenting an agent with a sequence of tasks with gradually increasing complexity, allowing the agent to eventually solve the initially given sparse reward task.

Due to the formal similarity between sparse rewards and trajectory feedback, both involving a single reward signal after a sequence, these methods can be directly applied to RLTF. However, the key difference lies in the nature of the rewards: sparse rewards typically focus on specific key points, making the reward Markovian, whereas RLTF evaluates the entire trajectory, indicating a Non-Markovian reward. Furthermore, sparse reward strategies address the exploration-exploitation trade-off inherent in sparse rewards, while RLTF focuses on distributing the evaluation of the entire trajectory across specific state-action pairs to guide learning. Thus, while these methods are applicable, they do not address the unique challenges posed by bagged rewards, as they fail to account for the reward structure within the bags. Therefore, they are not ideal solutions for RLTF (Gangwani et al., 2020; Ren et al., 2021).

In Fig. 6, we present additional results comparing our method with sparse reward baselines. Shaping (Tessler et al., 2019; Hu et al., 2020) and LIRPG (Zheng et al., 2018) are as described in the main paper. The Random Network Distillation (RND) (Burda et al., 2019b) method introduces an exploration bonus for deep reinforcement learning by measuring the prediction error of a randomly initialized neural network, which is a widely recognized and utilized approach for addressing sparse reward challenges. We include it as a baseline in our comparison of sparse reward methods for a more comprehensive evaluation.

These results demonstrate that, despite the superficial similarity between sparse rewards and bagged rewards, the internal structure of the rewards differs, leading to suboptimal performance of sparse reward baselines

Table 4: Performance comparison across reward bag with various-length configurations over 3 trials. In this table, "Short" refers to bags with lengths varying from 25 to 200, and "Long" denotes the setting with bag lengths from 100 to 500. The best and comparable methods based on the paired t-test at the significance level 5% are highlighted in boldface.

| Bag Setting | Environment | SAC | IRCR | RRD | HC | Shaping | LIRPG | RBT(ours) |
|---|---|---|---|---|---|---|---|---|
| Short | Ant-v2 | 3.21 (1.97) | 269.12 (224.66) | 2661.56 (1675.65) | 20.03 (46.35) | -596.02 (178.75) | -1407.15 (504.59) | **5359.85 (129.28)** |
| | Hopper-v2 | 286.53 (72.51) | 3275.05 (44.68) | 2508.76 (690.07) | 474.21 (66.93) | 1358.92 (640.58) | 287.38 (114.65) | **3433.06 (96.89)** |
| | HalfCheetah-v2 | 5.92 (23.06) | 10480.33 (202.27) | **10382.80 (516.85)** | 3982.43 (433.75) | 2010.72 (1085.36) | 1225.35 (162.88) | **11073.88 (181.43)** |
| | Walker2d-v2 | 222.21 (66.72) | 3840.30 (666.24) | 3999.14 (561.49) | 348.36 (174.28) | 926.49 (100.37) | 328.97 (107.69) | **5198.09 (225.50)** |
| Long | Ant-v2 | -115.10 (138.93) | 215.38 (92.94) | 2600.64 (1229.27) | -0.53 (5.07) | -460.51 (258.06) | -2552.99 (419.44) | **4897.50 (292.93)** |
| | Hopper-v2 | 360.36 (118.06) | 3015.96 (408.08) | 3089.52 (433.23) | 652.28 (92.41) | 729.28 (338.56) | 325.84 (50.25) | **3447.64 (83.02)** |
| | HalfCheetah-v2 | -115.56 (35.01) | 5944.17 (3421.46) | 8591.38 (1048.08) | 4563.99 (568.19) | -268.64 (114.37) | -571.35 (374.64) | **10880.76 (441.65)** |
| | Walker2d-v2 | 251.11 (144.24) | 3397.93 (682.19) | 4221.37 (282.75) | 497.27 (134.06) | 779.43 (219.75) | 284.27 (7.54) | **4979.07 (166.95)** |

in bagged reward environments. Conversely, as shown in the Section Compare with SOTA Methods of the main paper, our method performs well even in sparse reward environments, surpassing baselines specifically designed for sparse reward scenarios. This highlights the broad applicability and adaptability of our approach across different reward settings.

## C   Additional Experimental Results

This section provides further analysis and insights through additional experiments to complement the main findings presented in our main paper.

### C.1   Experimental Result of Various-Length Reward Bags

In Table 4, we present the experiment results on various-length reward bags. The experiment depict in the table showcases the results of various methods applied across different environments with varying bag lengths of rewards, where bags are one next to another as in the definition of RLBR. This experiment reveals that longer bags tend to degrade the performance of most methods. However, our RBT method appears to be less sensitive to changes in bag length, maintaining robust performance even when the bag length is equal to the full trajectory. This result aligns with the result in the Section Compare with SOTA Methods in our main paper.

### C.2   Preference on Handling Complex Delayed Reward Structures

The previous experiments were based on the assumption that a bagged reward is the sum of immediate rewards, which aligns with prior works (Gangwani et al., 2020; Ren et al., 2021; Zhang et al., 2023). However, our approach does not make this assumption, indicating that the proposed method can effectively handle more complex reward structures. In this section, we experimentally validate this claim. Specifically, we included the following complex delayed reward structures computed over $B_{i,n_i}$:

- **SumSquare**: The delayed reward is the sum of squared step rewards, placing more emphasis on larger rewards: $R_{\text{co}} = \sum_{t=i}^{i+n_i-1} \text{abs}(r_t) \cdot r_t$.

- **SquareSum**: The delayed reward is the square of the sum of step rewards, highlighting overall sequence performance: $R_{\text{co}} = \text{abs}\left( \sum_{t=i}^{i+n_i-1} r_t \right) \cdot \left( \sum_{t=i}^{i+n_i-1} r_t \right)$.

- **Max**: The delayed reward is a softmax-weighted sum of step rewards, giving more attention to the highest rewards: $R_{\text{co}} = \sum_{t=i}^{i+n_i-1} \frac{n_i \cdot e^{\beta r_t}}{\sum_{t'=i}^{i+n_i-1} e^{\beta r_{t'}}} \cdot r_t$,

  where $\beta$ is a scaling parameter that controls the sharpness of the softmax distribution.

The normalized scores shown in the figures are computed by scaling the obtained rewards against the maximum possible rewards achievable in these bagged reward environments. For SumSquare (see Figure 7), our method consistently outperforms baseline methods across different environments, maintaining strong performance even as the bag length increases. In the SquareSum setting (see Figure 8), the structure of bagged reward is more complex, as it emphasizes the cumulative effect of multiple steps rather than focusing on individual large rewards, making it more difficult to attribute rewards to specific actions. Despite this complexity, our method still holds a clear advantage over most baselines, even as performance slightly decreases with longer bag lengths. In the Max setting (see Figure 9), where rewards are sparse and only a few key steps contribute to the overall outcome, our method manages to learn effectively, showing that it can still identify and focus on the most critical instances in the sequence, even with limited reward signals. Under these conditions, methods relying on the sum-form delayed rewards assumption experience a significant drop in performance. This indicates that these methods are heavily dependent on this assumption. It is also worth noting that the HC method does not rely on this assumption, but its effectiveness is limited to shorter delays. As the length of bag increases, its performance drops sharply.

Overall, our proposed method shows a strong ability to handle various bagged reward structures and adapt to increasing lengths, outperforming baseline methods in most environments. Moreover, these results highlight the importance of further exploring how to learn reward models and train policies in environments with complex bagged reward structure, making it worthy of deeper investigation.

## C.3 Architecture Sensitivity

Fig. 10 illustrates the sensitivity of our architecture to different input sequence lengths during training (Seq len) and prediction lengths during reward relabeling (Relabel len) in the Ant-v2 environment. The learning curves, based on three independent runs with random initialization, show how varying these parameters affects the performance of the agent.

Although the configuration highlighted by the red box (Seq len of 500 and Relabel len of 100) demonstrates the best performance, the results also show that our proposed model is capable of learning effectively across various other configurations. This analysis underscores the importance of tuning input sequence length during training and prediction length during reward relabeling for optimal performance, while also demonstrating the ability of the RBT model to learn under different parameter settings, showcasing its flexibility and effectiveness in reinforcement learning tasks.

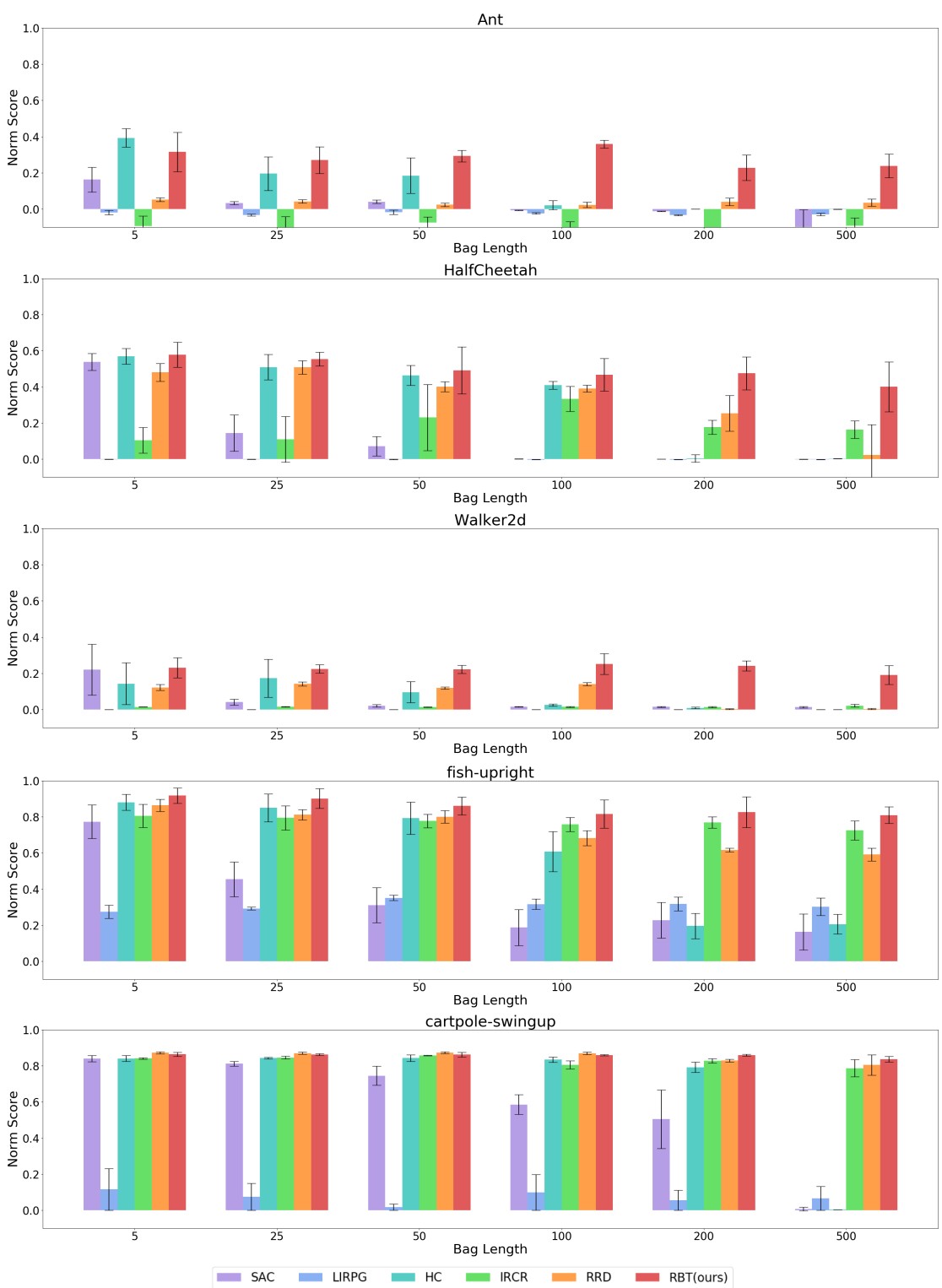

Figure 7: Performance comparison on the SumSquare reward structure with six different bag length settings (5, 25, 50, 100, 200, and 500). The mean and standard deviation are reported over six trials with different random seeds, measured across a total of 1e6 time steps.

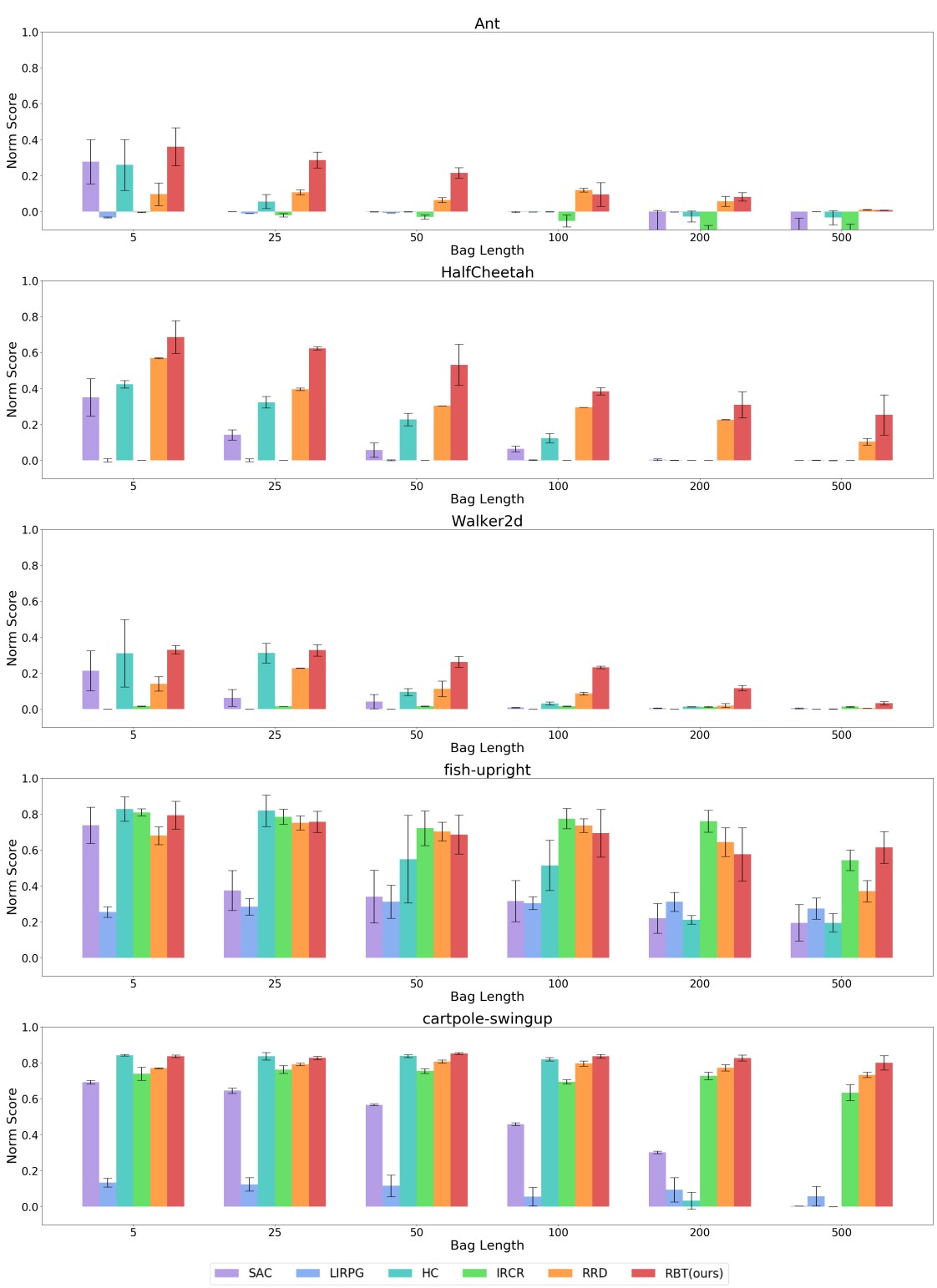

Figure 8: Performance comparison on the SquareSum reward structure with six different bag length settings (5, 25, 50, 100, 200, and 500). The mean and standard deviation are reported over six trials with different random seeds, measured across a total of 1e6 time steps.

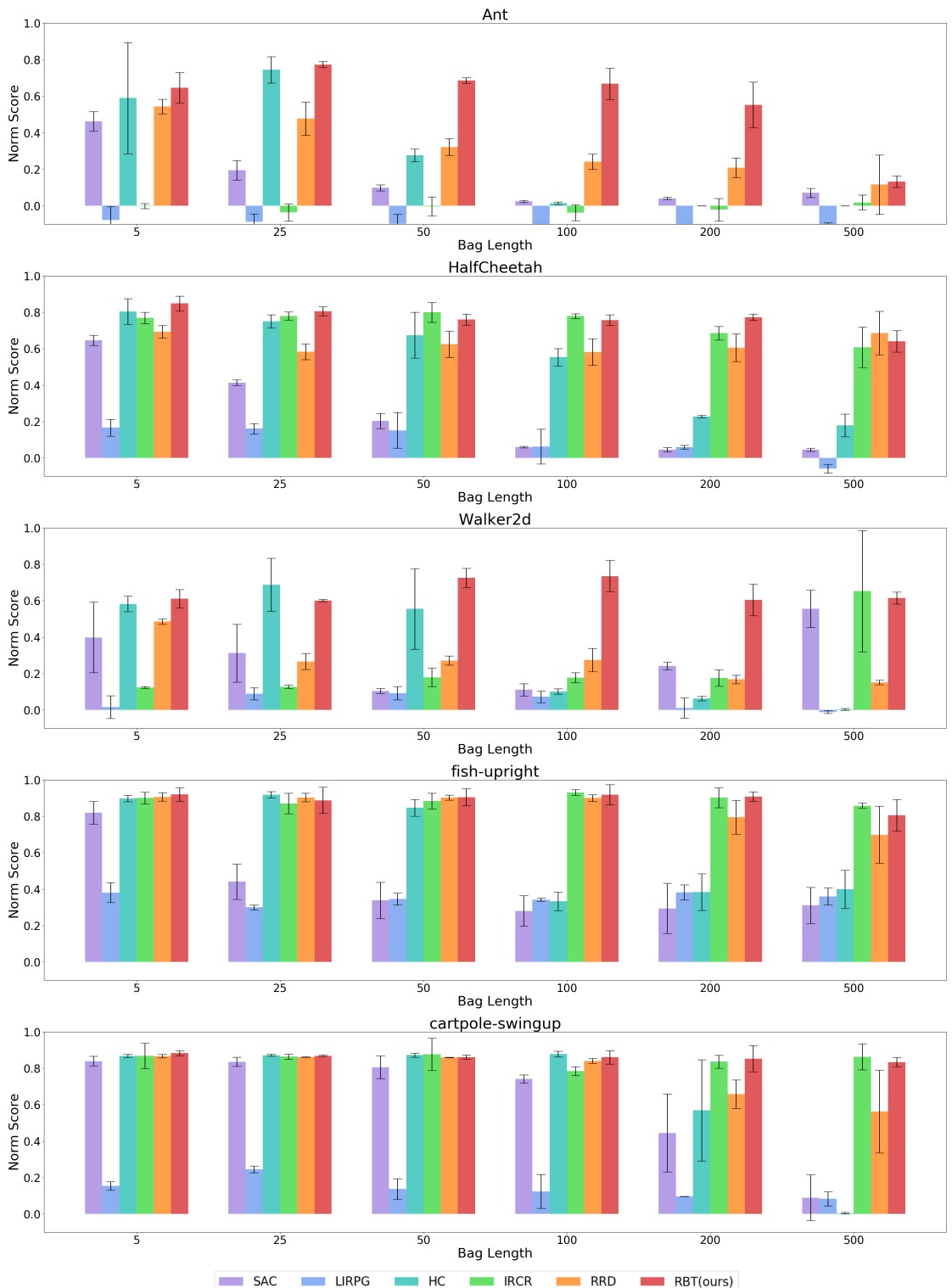

Figure 9: Performance comparison on the Max reward structure with six different bag length settings (5, 25, 50, 100, 200, and 500). The mean and standard deviation are reported over six trials with different random seeds, measured across a total of 1e6 time steps.

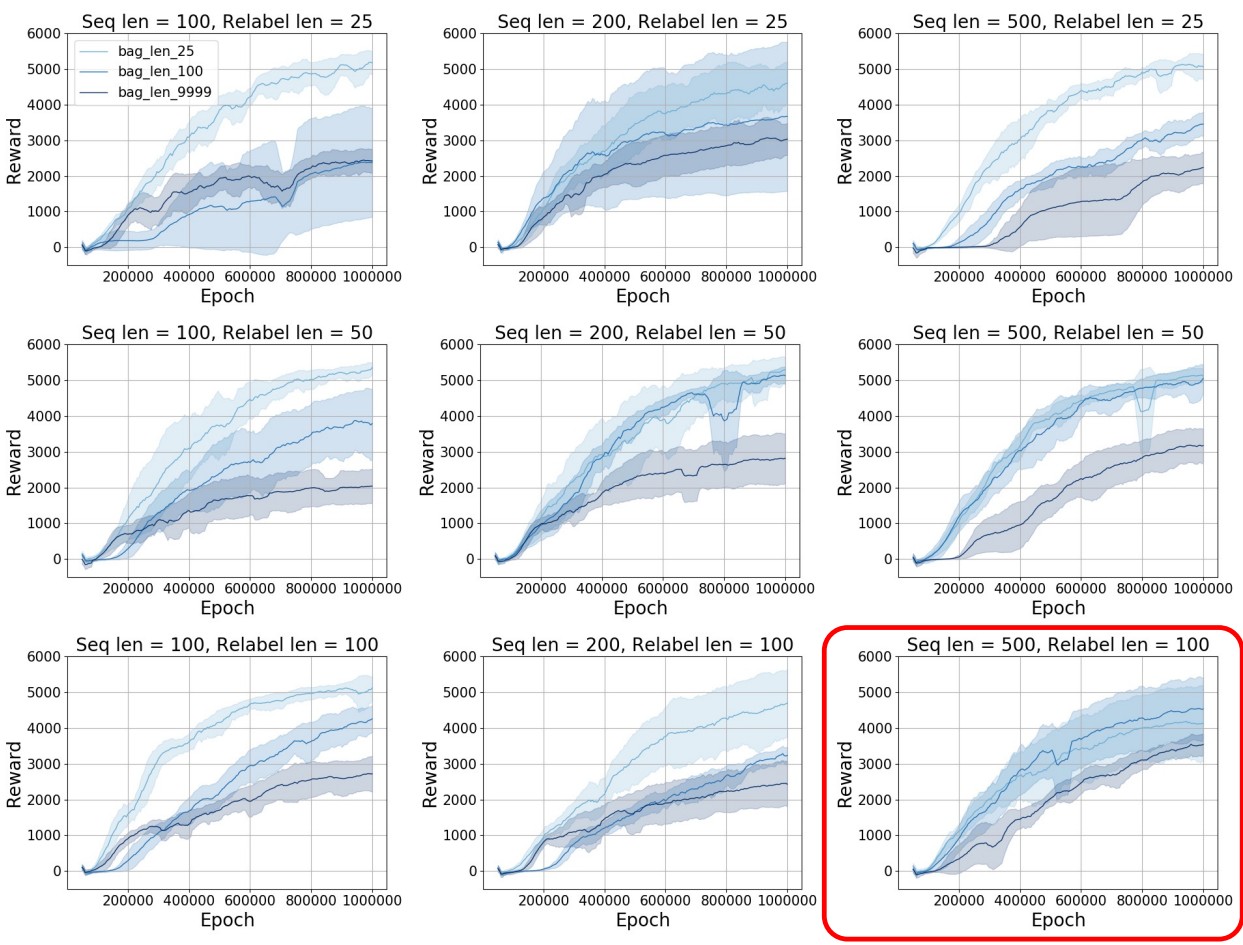

Figure 10: Learning curves on Ant-v2 with different length of input sequence in training (Seq len) and predict length during relabeling process (Relabel len), based on 3 independent runs with random initialization. Within each of the smaller graphs, the curves represent results from experiments with different bag lengths. Specifically, there are three bag lengths evaluated: 25, 100, and what is labeled as 9999, which we interpret as a proxy for trajectory feedback. The graph highlighted by the red box indicates our chosen parameter setting for the experiment, which is a input sequence length of 100 and a predict length of 500.

## C.4 Final Training Losses

Table 5: Final training losses at the trajectory level on the Ant-v2 environment with different bag lengths, showing the mean and standard deviation over three random seeds.

| Bag Length ($n$) | 5 | 25 | 50 | 100 | 200 | 500 | 9999 |
|---|---|---|---|---|---|---|---|
| State Prediction Loss | $0.003 \pm 0.002$ | $0.023 \pm 0.001$ | $0.031 \pm 0.007$ | $0.032 \pm 0.013$ | $0.056 \pm 0.036$ | $0.053 \pm 0.027$ | $0.032 \pm 0.011$ |
| Reward Prediction Loss | $0.078 \pm 0.003$ | $0.174 \pm 0.008$ | $0.310 \pm 0.049$ | $0.769 \pm 0.061$ | $1.342 \pm 0.146$ | $3.066 \pm 0.529$ | $6.195 \pm 0.406$ |
| Composite Loss | $0.081 \pm 0.001$ | $0.196 \pm 0.008$ | $0.341 \pm 0.056$ | $0.801 \pm 0.068$ | $1.399 \pm 0.116$ | $3.120 \pm 0.520$ | $6.227 \pm 0.396$ |

The training losses vary with different bag lengths, reflecting how the model's learning process is influenced by the difficulty of reward redistribution and state prediction. To ensure a fair comparison, the estimated reward prediction loss, estimated state prediction loss, and estimated composite loss presented in Table 5 are all computed based on the entire trajectory. This means that for each trajectory, we first sum the reward prediction loss over all reward bags and the state transition loss over all state-action transitions. The overall composite loss is then obtained by summing these two aggregated losses. The results show that the state prediction loss remains relatively stable across different bag lengths, suggesting that the model effectively learns environmental dynamics with minimal impact from the bag size. In contrast, the reward prediction loss increases significantly with longer bag lengths. This growth is not only due to the increased difficulty of redistributing rewards over a larger set of instances but also stems from the inherent squared error accumulation in Equation 5. Even if the per-instance prediction error remains constant, the total loss scales with bag length because the squared error accumulates over all instances within the bag. For example, assuming a constant instance-level error $e$ and trajectory length 100, a short bag of length $n = 5$ results in an accumulated squared error of $(100/5) \times (5 \times e)^2 = 500 \times e^2$, whereas a long bag of length $n = 50$ leads to a much larger accumulated error of $(100/50) \times (50 \times e)^2 = 5000 \times e^2$. This explains why longer bags naturally lead to higher reward prediction loss, even when individual instance errors remain the same. Consequently, since composite loss is a combination of state prediction loss and reward prediction loss, it is primarily driven by the increasing reward prediction loss at larger bag lengths. This suggests that the main challenge in learning under bagged rewards lies in accurate reward redistribution when fewer direct supervisory signals are available. Future improvements could focus on mitigating the impact of large bag lengths on reward prediction, such as incorporating regularization techniques, hierarchical reward redistribution, or adaptive weighting in composite loss to improve overall learning efficiency.

Table 6: Normalized reward prediction losses in the Ant-v2 environment for different bag lengths, computed as the mean squared error between the sum of predicted instance-level rewards and the bagged reward, normalized by the square of the bag length. The table reports the mean over three random seeds.

| Bag Length ($n$) | 5 | 25 | 50 | 100 | 200 | 500 | 9999 |
|---|---|---|---|---|---|---|---|
| Normalized Loss | $1.56 \times 10^{-5}$ | $6.96 \times 10^{-6}$ | $6.2 \times 10^{-6}$ | $7.69 \times 10^{-6}$ | $6.71 \times 10^{-6}$ | $6.13 \times 10^{-6}$ | $6.20 \times 10^{-6}$ |

We also present normalized reward prediction losses in Table 6. The normalization is performed as:

$$\text{Normalized Loss} = \frac{\mathcal{L}_r(B)}{n^2}, \tag{8}$$

where $\mathcal{L}_r(B)$ represents the reward prediction loss for a single bag, computed as the squared difference between the sum of predicted instance-level rewards and the bagged reward. This normalization is motivated by the fact that $\mathcal{L}_r(B)$ is a mean squared loss, meaning that its value naturally scales with $n^2$ due to the summation inside the squared term. By dividing by $n^2$, we obtain a loss that is comparable across different bag lengths. This adjustment allows for a fair per-instance comparison across different bag lengths. Interestingly, shorter bag lengths tend to have a higher normalized loss. Since the loss function aggregates squared differences within each bag, a possible reason for this is that having more bags means errors are accumulated and squared more frequently, leading to an overall increase in per-instance loss.

Theorem 1 establishes that if the sum of redistributed rewards within each bag perfectly matches the bagged reward, the optimal policies in the BRMDP and the redistributed reward MDP are identical. However, in practical settings, learning a reward model introduces approximation errors, meaning the redistributed rewards may not sum exactly to the bagged rewards. A small non-zero loss in reward redistribution leads to slight deviations in cumulative rewards along trajectories. While such deviations exist, our results in Fig. 3 suggest that the learned policies still achieve performance comparable to those optimized with true rewards, indicating that the impact of small redistribution errors is limited. However, if the discrepancies were larger, they could potentially distort policy learning and lead to different optimization outcomes. In such cases, additional techniques, such as methods for enhancing policy robustness to noisy rewards (Wang et al., 2020), may be required to mitigate these effects. While our theoretical analysis assumes perfect reward redistribution, this assumption holds approximately in practical implementations where the learned reward model achieves low prediction error. Investigating the precise impact of redistribution errors on policy learning is an important direction for future work.

### C.5 Runtime and Computational Overhead

Table 7: Training runtime (in hours) for different methods.

| Method | SAC | IRCR | Shaping | LIRPG | RRD | HC | RBT (ours) |
|---|---|---|---|---|---|---|---|
| **Runtime (hours)** | 1 | 1 | 1 | 8 | 21 | 30 | 22 |

The Table 7 presents the training runtime for different methods on an NVIDIA GeForce RTX 2080 Ti GPU cluster with 8GB of memory. Only RBT (ours) and RRD utilize a reward model, which introduces additional computational overhead, resulting in longer training times compared to methods that do not rely on a reward model. HC modifies the value function to adapt to sequence-level rewards, which also increases training time.

Notably, the primary factor contributing to the runtime of our proposed RBT method is not the training of the model itself but rather the relabeling of rewards for samples in the replay buffer. This process adds significant computational overhead, making it an area for potential optimization. A potential improvement is to use prioritized experience replay, where more informative samples are prioritized for relabeling instead of treating all samples equally, thereby improving efficiency while maintaining learning quality. Another approach is to implement a cached reward update mechanism, where only a subset of samples is updated per iteration rather than recomputing rewards for all samples, significantly reducing computational costs. By incorporating these optimizations, the computational efficiency of RBT has the potential to be significantly improved without compromising performance. By incorporating these optimizations, the computational efficiency of RBT has the potential to be significantly improved without compromising performance.

