# OpenReview forum: "Reinforcement Learning from Bagged Reward"
_TMLR — Accepted by TMLR_

### Review · Reviewer_u6fd · 2025-02-20

**Summary Of Contributions:**

The submission considers decision-making where the reward is sparse and defined on trajectories instead of one single state-action pair. To tackle this problem, the authors first propose to formulate the problem as an MDP with bagged reward and connect this formulation withe an ordinary MDP but with properly redistributed reward. In other words, the key is to properly redistribute the reward. Therefore, the authors propose a novel reward redistribution method built transformers.

Both theoretical and empirical analysis are provided, demonstrating the advantages of the proposed method.

**Audience:**

Yes

**Claims And Evidence:**

Yes

**Requested Changes:**

I do not have any suggested adjustments.

**Strengths And Weaknesses:**

The submission is well-written and easy to follow.

The results are thorough with both theoretical and thorough empirical analysis. The improvement of the proposed method is well demonstrated.

According to my knowledge, the proposed method is novel and tackles an important problem, where the reward is defined on a trajectory in a non-Markov and not-necessarily additive manner.

---

> ### Author Response · Authors · 2025-02-20
>
> Dear Reviewer u6fd,
>
> We sincerely appreciate your positive comments on the clarity of our writing, the thoroughness of our theoretical and empirical analysis, and the novelty of our proposed method.
>
> Thank you for your time and effort in reviewing our submission.
>
> Best,
>
> Authors

---

### Review · Reviewer_voSh · 2025-02-25

**Summary Of Contributions:**

The paper proposes bagged-reward MDPs, which can be viewed as an advanced-reward shaping approach. In particular, the authors propose a transformer to distribute bagged rewards among the chosen action state pairs. The approach is evaluated against other more naive ways to distribute the reward and for different bag time horizons on MuJoCo tasks. This shows the clear superiority of using the transformer to reshape over the other approaches. Additionally, the authors perform an ablation study for their transformer architecture demonstrating that all proposed components (bi-directional attention and state estimation) are necessary for the best performance.

The paper is well written and the figures and tables support the understanding well.

**Audience:**

Yes

**Claims And Evidence:**

Yes

**Requested Changes:**

- Theorem 1:
  - My understanding of the theorem is that if the sum of the redistributed reward of one bag is equivalent to the bag reward, the MDP and BRMDP are equivalent. If that is correct, this is not clear from the theorem, please rephrase or explain
  - I could not follow the proof logic - if my assumption is correct the last equation in the proof would suffice in my opinion. I would include the shorter proof in the main text or give an abbreviated proof there.
  - Also the wording is not identical for Theorem 1 in the text and in Appendix A, this should be corrected

- Limitation section - I see several undiscussed points that could be answered by a limitation section, additional experiments or revised related work depending on your preference:
  - What training is necessary for the transformer (how long, how many trajectories are used, what are the final losses $\mathcal{L}_{bag}$, $\mathcal{L}\_{r}$ and $\mathcal{L}\_{s}$ ..)? Do you train a new one for each task?
  - How do you ensure that the sum of the redistributed reward is equal to the bagged reward?
  - Why did you not compare to vanilla SAC in your experiments? This would be the minimum that should be achievable and a good reference for all other approaches.
  - While your approach addresses many issues of reward engineering, it still assumes a reward function (sparse or dense) is available. This can still be difficult to obtain in the first place. What do you think about extending your approach with rewards that are automatically generated from task specifications [1-3]?

- The contribution paragraph is not too clear. Pleas separate contributions and your solution approach more clearly.
- The origin of the bagged rewards was not too clearly introduced. I would appreciate a revised experimental setup that more clearly states what is given and what is calculated.

References:

[1]  Rodrigo Toro Icarte, Toryn Q. Klassen, Richard Valenzano, and Sheila A. McIlraith. 2022. Reward Machines: Exploiting Reward Function Structure in Reinforcement Learning. J. Artif. Int. Res. 73 (May 2022). https://doi.org/10.1613/jair.1.12440

[2]  LTL and Beyond: Formal Languages for Reward Function Specification in Reinforcement Learning. Alberto Camacho, Rodrigo Toro Icarte, Toryn Q. Klassen, Richard Valenzano, Sheila A. McIlraith. Proceedings of the Twenty-Eighth International Joint Conference on Artificial Intelligence Understanding Intelligence and Human-level AI in the New Machine Learning era. Pages 6065-6073. https://doi.org/10.24963/ijcai.2019/840

[3] Le, X. B., Wagner, D., Witzman, L., Rabinovich, A., & Ong, L. Reinforcement Learning with LTL and $\omega $-Regular Objectives via Optimality-Preserving Translation to Average Rewards. In The Thirty-eighth Annual Conference on Neural Information Processing Systems.

**Strengths And Weaknesses:**

Strengths:
- The experiments compare many baselines and are well-executed
- The paper addresses a central issue of RL, and the proposed solution seems to be superior for dense and spare reward environments, which is an exciting result.

Weaknesses:
- The theorem and respective proof are not clear.
- The paper lacks a limitation section that discusses caveats of the results and method.

---

> ### Author Response · Authors · 2025-03-04
>
> Thank you for the detailed review! We have modified the paper according to your suggestions, with the revised parts highlighted in blue. Below are our responses to your questions.
>
> **Q1: The MDP and BRMDP are equivalent.**
>
> A1: Your understanding is quite close. This theorem specifically states that if the bagged reward is redistributed appropriately, the optimal policies learned in the MDP and BRMDP will be the same, rather than implying that the MDP and BRMDP are structurally equivalent. To clarify this point, we have rephrased the theorem accordingly in our revision.
>
> **Q2: Include the shorter proof in the main text or give an abbreviated proof in appendix.**
>
> A2: Thank you for your suggestion. We have removed the proof from the appendix and included a concise version in the main text.
>
> **Q3: Discussion on runtime.**
>
> A3: The table presents the training runtime for different methods on an NVIDIA GeForce RTX 2080 Ti GPU cluster with 8GB of memory. Notably, among all methods, only our approach (RBT) and RRD utilize a reward model, leading to similar training times. HC modifies the value function to adapt to sequence-level rewards, which results in a longer runtime.
> The extended training time for RBT is primarily due to the reward relabeling process, which involves updating the assigned rewards in the replay buffer based on the learned reward model. This additional computational step increases the overall training time compared to methods that do not require reward relabeling.
> In the updated version of the paper, we have included the runtime results along with a discussion on potential methods for improving computational efficiency in **Appendix C.5**. Additionally, we have expanded the **Limitations** section to further address this aspect.
>
> |Method|SAC|IRCR|Shaping|LIRPG|RRD|HC|RBT (ours)|
> |-|-|-|-|-|-|-|-|
> |**Runtime (hours)**|1|1|1|8| 21|30|22|
>
> **Q4: Required sample size and trajectory length during training.**
>
> A4: We train the Transformer using 1e6 state-action pairs per task. Each trajectory has a maximum length of 1000 steps, and we maintain consistent settings across all environments and baseline methods to ensure fair comparisons. We have emphasized this part in **Section 5.1.1**.
>
> **Q5: Final training loss.**
>
> A5: The final training losses vary depending on the bag length, with the reward prediction loss increasing as the bag length grows, while the state prediction loss remains relatively stable. This indicates that longer bag lengths introduce more challenges in accurately redistributing rewards. We have added the corresponding results and discussion in **Appendix C.4** for further reference.
>
> **Q6: Do you train a new model for each task?**
>
> A6: Yes, we train a separate Transformer model for each task to account for task-specific reward structures and environmental dynamics.
>
> **Q7: How to ensure the sum of redistributed rewards equals the bagged reward?**
>
> A7: We ensure that the sum of the redistributed rewards equals the bagged reward through the $\mathcal{L}_{\textrm{r}}$ introduced in Section 4.1.1. This loss function explicitly enforces the consistency between the sum of redistributed rewards and the original bagged reward. We have revised our original explanation in this section to make it clearer.
>
> **Q8: Comparison with vanilla SAC in experiments.**
>
> A8: In our baseline, the "SAC" method refers to applying vanilla SAC to trajectories using the original bagged reward. Therefore, our experiments already include a comparison with vanilla SAC as a reference point.
>
> **Q9: What do you think about extending your approach with rewards that are automatically generated from task specifications?**
>
> A9: Thank you for your insightful question. Our approach indeed assumes that a reward function is available, which can be a limitation when reward design is challenging. Extending our method to automatically generated rewards from task specifications is an interesting and promising direction.
>
> One potential way to extend our approach is to integrate task-specification-driven reward generation with our reward redistribution framework. For instance, automatically generated rewards could serve as the initial bagged rewards, which our method could then redistribute at the instance level. This would create a fully automated pipeline, from task specification to policy learning, without requiring manually defined rewards.
> We appreciate the references and will explore this direction further in future work!

---

> ### Author Response · Authors · 2025-03-04
>
> **Q10: Rewriting the contribution section.**
>
> A10: Thank you for your suggestion. We have rewritten the **final paragraph of Section 1** to clarify the contributions and distinguish them from our solution approach. We hope this revision makes our contributions sufficiently clear.
>
> **Q11: Description of bagged rewards in the experimental setup.**
>
> A11: Thank you for your suggestion. We have revised **Section 5.1.1** to clarify the origin of the bagged rewards.
>
> We hope our responses have addressed your concerns. Please feel free to reach out if you have any further questions. We would be happy to discuss them.

---

> ### Comment · Reviewer_voSh · 2025-03-12
> **Reply to authors response**
>
> Thanks for your thorough answers and the revised paper. Especially, the revised proof is much clearer.
>
> I have a few questions remaining:
> - I think my comment was not very clear: by "vanilla" SAC, I meant without the bagged reward and with the standard reward definition. Comparing this with your existing SAC with bagged rewards would show how much improvement can be gained from adding the bagged rewards for a specific task.
> - A follow-up on the loss function $\mathcal{L}_r$ defined in Eq. 5: When the loss is almost zero, it is ensured that the reward and bagged reward are equal. In Table 5 for the Ant environment, the loss is only sometimes close to zero and especially for long bag lengths this error seems to grow significantly. This means, in practice, the assumption that the re-distribution is correctly executed by the transformer so that Theorem 1 holds is not true. Is this deviation from theory visible in the trained policies? Here standard SAC (with standard reward function) would be again the baseline to compare with.

---

> ### Author Response · Authors · 2025-03-14
>
> Thank you for your question. Below are our responses to these two questions, and we hope our explanations address your concerns. We have also updated the manuscript for your reference.
>
> **Q1: Vanilla SAC baseline results.**
>
> A1: Thank you for your explanation! We have updated Figure 3. The **dashed lines in Figure 3** correspond to the results of vanilla SAC. In addition, we have also included the corresponding description in **Section 5.4.1 Experimental Results**.
>
> **Q2: Does the increasing loss for longer bag lengths in Table 5 indicate that the reward redistribution is not perfectly executed, potentially affecting the validity of Theorem 1 and the trained policies?**
>
> A2: Thank you for your insightful question.
> In Table 5, the estimated reward prediction loss, estimated state prediction loss, and estimated composite loss are all computed over the entire trajectory to ensure a fair comparison. This means that for each trajectory, we first sum the reward prediction loss over all reward bags and the state transition loss over all state-action transitions. The overall composite loss is then obtained by summing these two aggregated losses.
> We have included this description in **Appendix C.4**.
>
> The increasing loss for longer bag lengths is expected due to the inherent nature of the loss function. Even if the true and predicted instance-level rewards for short and long bags are identical, the squared error accumulation in Equation (5) causes the overall loss to be larger for longer bags.
>
> For example, suppose the instance-level error is $0.01$ and the total trajectory length is $100$. Consider two scenarios:
> - Short Bag ($n = 5$): The accumulated squared error is $(100/5) * (0.01 * 5)^2 = 0.05$.
> - Long Bag ($n = 50$): The accumulated squared error is $(100/50) * (0.01 * 50)^2 = 0.5$.
>
> This explains why longer bags naturally lead to higher loss values, even if the per-instance error remains small. This means, while some deviation exists, the redistributed rewards still closely approximate the original bagged rewards.
>
> We greatly appreciate this observation, as reducing this deviation could further refine the method. A possible improvement could involve scaling the loss function adaptively to normalize the contribution of different bag lengths or exploring contrastive loss formulations that focus on relative consistency rather than absolute magnitude.
> We hope this explanation clarifies the concern. This discussion is also included in **Appendix C.4**.
> Please let us know if any further details are needed!

---

> ### Comment · Reviewer_voSh · 2025-03-17
> **Second reply to authors**
>
> Thanks a lot for your detailed answers. I would appreciate it if you adapt your paper with respect to the following two comments:
>
> After looking at Fig. 3, it seems like for "quadruped-walk," your approach outperforms the vanilla SAC baseline with bag length 1, and for the other environments, your approach approximately matches the vanilla SAC with bag length 1. Can you provide an intuition for this? Does this lead to insights on when to use your approach and when to use vanilla SAC?
>
> Thanks for explaining Tab. 5 in more detail here and in the text. I would consider normalizing the reported errors with the bag length to make the table more easily comprehendable. Also, please clarify how any non-zero loss affects your statement in the theorem.

---

> > ### Author Response · Authors · 2025-03-17
> >
> > Thank you for your thoughtful feedback and suggestions. We appreciate the opportunity to clarify these points and have incorporated additional analysis to address your concerns. Below, we provide detailed responses to each question, and we hope our explanations help resolve any uncertainties.
> >
> > **Q1: After looking at Fig. 3, it seems like for "quadruped-walk," your approach outperforms the vanilla SAC baseline with bag length 1, and for the other environments, your approach approximately matches the vanilla SAC with bag length 1. Can you provide an intuition for this? Does this lead to insights on when to use your approach and when to use vanilla SAC?**
> >
> > A1: Thank you for your insightful question. Regarding the "quadruped-walk" results, the lower mean performance of the vanilla SAC baseline with a bag length of 1 is primarily due to the variance across different random seeds. Specifically, among the six evaluation seeds, one yielded a significantly lower return compared to the others: 954.14, 918.84, 491.60, 956.44, 954.04, 941.32.
> > The lower-performing seed (491.60) significantly reduced the average return of SAC, making our approach appear to outperform it. To provide a clearer interpretation of the results, we have included the standard deviation information in Fig.3 for reference.
> >
> > One possible explanation for this outlier in the SAC results is the sensitivity of the "quadruped-walk" environment to exploration noise. This task requires precise coordination of multiple joints over a sequence of steps, making early-stage exploration crucial. If an agent initially learns a suboptimal gait pattern or fails to stabilize its movements, it may struggle to recover, leading to significantly lower final performance. The presence of one poorly performing seed suggests that small differences in early exploration trajectories can have a lasting impact on policy learning in this environment.
> > It is also worth noting that our approach is not entirely immune to this issue. For example, in the trajectory feedback setting (bag length 9999) of "quadruped-walk", the performance variance is also relatively high. This suggests that while reward redistribution can help stabilize learning in some cases, environments with high exploration sensitivity can still exhibit significant performance fluctuations across seeds.
> >
> > Based on results across multiple environments, we observe that our approach is particularly beneficial in scenarios where reward signals are sparse, delayed, or require better credit assignment. In such cases, redistributing rewards over sequences helps stabilize learning and improve performance. However, in environments where per-step rewards are already well-defined and provide strong supervision, vanilla SAC may already be optimal, and reward redistribution may offer limited additional benefits.

---

> ### Author Response · Authors · 2025-03-17
>
> **Q2: I would consider normalizing the reported errors with the bag length to make the table more easily comprehendable. Also, please clarify how any non-zero loss affects your statement in the theorem.**
>
> A2: Thank you for your suggestion. We have added **Table 6 in Appendix C.4**, which presents the normalized reward prediction loss to improve clarity. From this table, we can observe that the loss per instance remains relatively small, suggesting that the overall redistribution error is limited. We have also included a discussion on these results to provide further insights in **Appendix C.4**.
> At the same time, we have retained Table 5 to examine the relationship between trajectory-level reward prediction loss, state prediction loss, and composite loss.
>
> In Theorem 1, we assume that the sum of redistributed rewards within each bag exactly matches the bagged reward. This assumption is crucial for establishing the equivalence between the optimal policies in BRMDP and the redistributed reward MDP. However, in practical training scenarios, the learned reward model may introduce small errors.
>
> A small non-zero loss in reward redistribution leads to slight deviations in cumulative rewards along trajectories. While such deviations exist, our results in Fig.3 suggest that the learned policies still achieve performance comparable to those optimized with true rewards, indicating that the impact of small redistribution errors is limited in practice.
> However, if the discrepancies were larger, they could potentially distort policy learning and lead to different optimization outcomes. In such cases, additional techniques, such as methods for enhancing policy robustness to noisy rewards [1], may be necessary to mitigate these effects.
>
> While our theoretical analysis assumes perfect reward redistribution, this assumption holds approximately in practical implementations where the learned reward model achieves low prediction error. Investigating the precise impact of redistribution errors on policy learning remains an important direction for future work.
> We have incorporated this discussion into **Appendix C.4** for further clarity. We appreciate the reviewer for raising this valuable point.
>
> [1] Reinforcement learning with perturbed rewards.

---

### Review · Reviewer_BSYC · 2025-03-07

**Summary Of Contributions:**

This paper tackles the problem of learning policies which can only observe delayed bagged rewards instead of instantaneous timestep rewards. They learn to invert the bagging transform (i.e. go back to instantaneous rewards) using a Transformer. They then apply off the shelf RL algorithms to obtain a policy. They show that their method is able to tackle simple control problems in the Control Suite.

**Audience:**

Yes

**Claims And Evidence:**

Yes

**Requested Changes:**

Critical: (none)

Strengthen:
1. Discuss the limitations above more directly

**Strengths And Weaknesses:**

Overall this is a nicely written paper which tackles a clearly defined problem simply but well.
   1. The problem setting is well presented, the choices and limitations are clearly specified and experiments are of limited scope (i.e. Control Suite only) but well executed.
   2. The proposed Transformer, which does both state prediction alongside reward prediction, is rather standard but the combination of Causal + Bidirectional layers does seem to work well and beats baselines.
   3. I appreciated Figure 4, which demonstrated how well the transformer performed at predicting rewards, it was informative and clear.

My main issue with the paper is probably about generalizability and knowing if this would handle more complex domains.
   1. The bag partition function G is assumed to be part of the environment. Unfortunately that seems like a strong requirement, and I can imagine that lifting this would make the problem setting much more complex (i.e. one would have to infer bag transitions, etc)
   2. Most of the main text revolves around assuming that the bagged reward is the sum of intermediate rewards. This isn’t strictly necessary for their work though, as covered in Appendix D.2, but this makes this problem setting look a bit more simple than it really could be. It might be worth bringing this up earlier in the main text (e.g. when defining the bagged reward as the sum).
   3. There is no strict presentation of generalisation ability across games / when domain transfer of the Transformer is expected, as far as I could see? All results present policies+reward functions trained and evaluated on environments independently, which is fair but again limited in scope.
   4. How does the reward function transfer across environments? Do you expect it to?
   5. The Control Suite is a simple and very smooth environment. Rewards are cyclical and simple to predict, the state space is also simple. How does the method handle more complex environments with partial observability and very sharp reward transitions? What about when exploration is difficult?

---

> ### Author Response · Authors · 2025-03-14
>
> Thank you for your suggestion. Below are our responses to each suggestion. Based on your feedback, we have updated the manuscript, particularly the **Limitations section**. We hope our explanations address your concerns.
>
> **Q1: The bag partition function $\mathcal{G}$ is assumed to be part of the environment. Unfortunately that seems like a strong requirement, and I can imagine that lifting this would make the problem setting much more complex (i.e. one would have to infer bag transitions, etc)**
>
> A1: We adopt the assumption that the bag partition function G is part of the environment to simplify the problem formulation, as it naturally aligns with many real-world scenarios. In practice, bag structures are often naturally defined by the task, such as segments of a financial trading session, episodic decision-making in robotics, or structured feedback in recommendation systems. This makes the assumption practical in various domains.
>
> At the same time, we acknowledge that an unknown $\mathcal{G}$ represents a more general situation.
> If $\mathcal{G}$ were unknown, it would introduce additional complexity, requiring a method to infer bag transitions. This could be tackled by latent variable modeling [1, 2, 3] or attention-based mechanisms [4] that learn to segment sequences dynamically. We consider this an exciting future direction, but in this work, we focus on the fundamental problem under a given $\mathcal{G}$ to establish the theoretical foundation of reward redistribution in RLBR. We have included a discussion on this in the **Limitation section**.
>
> **Q2: Most of the main text revolves around assuming that the bagged reward is the sum of intermediate rewards. This isn’t strictly necessary for their work though, as covered in Appendix D.2, but this makes this problem setting look a bit more simple than it really could be. It might be worth bringing this up earlier in the main text (e.g. when defining the bagged reward as the sum).**
>
> A2: Thank you for your suggestion. To clarify, assuming that the bagged reward is the sum of instance-level rewards is not a fundamental part of the problem setting but rather a key assumption of our proposed method. This assumption ensures the equivalence of optimal policies in the BRMDP and the MDP after reward redistribution (Theorem 4.1) and provides a foundation for our reward redistribution method. As discussed in Appendix D.2, the bagged reward can also be defined through other aggregation mechanisms. In response to this concern, we have revised **Section 3** to explicitly clarify this distinction.
>
> **Q3: There is no strict presentation of generalization ability across games / when domain transfer of the Transformer is expected, as far as I could see? All results present policies+reward functions trained and evaluated on environments independently, which is fair but again limited in scope.**
>
> A3: We acknowledge that our current experiments focus on training and evaluating policies and reward redistribution functions within individual environments rather than across different environments. The primary goal of this work is to demonstrate the effectiveness of reward redistribution within a given environment before exploring domain transfer scenarios. We recognize that cross-environment generalization is an important future direction and have discussed it in the **Limitation section**.
>
> **Q4: How does the reward function transfer across environments? Do you expect it to?**
>
> A4: In principle, the learned reward redistribution model could be transferred across environments if the underlying reward structure remains similar. Pretraining on multiple environments or incorporating meta-learning techniques [5, 6, 7] could enhance generalization. Exploring reward function transfer across domains is an interesting future direction, and we have also discussed this in the **Limitation section**.

---

> ### Author Response · Authors · 2025-03-14
>
> **Q5: The Control Suite is a simple and very smooth environment. Rewards are cyclical and simple to predict, the state space is also simple. How does the method handle more complex environments with partial observability and very sharp reward transitions? What about when exploration is difficult?**
>
> A5: While the Control Suite is a relatively simple environment with smooth dynamics, it serves as a clean and controlled testbed to validate our theoretical framework before extending it to more complex settings. We will explore more challenging environments in future work.
>
> Regarding partial observability, our method leverages sequence-based modeling, such as Transformers, to capture long-term dependencies and infer missing information. However, in highly stochastic environments where key information is completely unobservable, techniques like belief state estimation [8, 9, 10] may be necessary.
>
> For sharp and sparse reward transitions, a straightforward approach is to use reward spikes as natural bag boundaries. In our experiments with sparse reward environments (Section 5.1 and Appendix B), we define bag boundaries based on the occurrence of reward signals. However, most existing sparse reward environments provide only trajectory-level rewards. In future work, we will consider exploring more complex environments with both sharp and sparse rewards. Another potential solution is adaptive bag partitioning, which could better align reward redistribution with critical decision points.
>
> In hard-exploration environments, our method focuses on credit assignment rather than exploration. While reward redistribution provides denser feedback, it does not directly address exploration challenges.
>
> We appreciate these insightful questions. We have made the necessary revisions in the paper and plan to explore these challenges in future work.
>
> [1] Hidden semi-Markov models.
> [2] Auto-encoding variational bayes.
> [3] Bayesian nonparametric hidden semi-Markov models.
> [4] Asformer: Transformer for action segmentation.
> [5] Model-agnostic meta-learning for fast adaptation of deep networks.
> [6] Fast context adaptation via meta-learning.
> [7] DARLA: Improving zero-shot transfer in reinforcement learning.
> [8] Planning and acting in partially observable stochastic domains.
> [9] A survey of POMDP solution techniques.
> [10] Deep variational reinforcement learning for POMDPs.

---

### Decision · Action_Editor_L2PZ · 2025-04-13

**Recommendation:** Accept with minor revision

**Comment:**

This paper addresses an important problem in reinforcement learning—credit assignment under delayed or sparse rewards—by introducing a novel reward redistribution method using bidirectional attention mechanisms. The reviewers unanimously recommended acceptance, citing the paper's strong theoretical foundation, clear presentation, and empirical rigor. However, they also raised some valid concerns about generalizability, scope, and assumptions. Below is a summary of these points and how they were addressed:

Generalizability Across Domains:
Reviewers noted that experiments were limited to simple environments (e.g., Control Suite) without demonstrating cross-domain generalization. The authors acknowledged this limitation and discussed it in the revised manuscript's "Limitations" section. They proposed future work on pretraining across environments or incorporating meta-learning techniques to enhance transferability.

Assumption of Known Bag Partition Function:
While the assumption simplifies the problem, reviewers suggested that lifting this constraint would make the approach more broadly applicable. The authors justified their choice by aligning it with practical scenarios but acknowledged this as a limitation and discussed potential extensions involving latent variable modeling.

Complex Environments:
Concerns were raised about how the method would perform in more complex settings with partial observability or sharp reward transitions. The authors responded by highlighting their method's adaptability through sequence-based modeling while acknowledging challenges in highly stochastic environments.

Scope of Reward Aggregation Assumption:
Reviewers pointed out that assuming bagged rewards are sums of intermediate rewards oversimplifies the problem setting. The authors clarified that this assumption is specific to their method but not fundamental to RLBR as a whole, revising Section 3 for better transparency.

These revisions adequately address the reviewers' concerns without requiring substantial additional work from the authors before publication.

The Action editor encourages the authors to connect with use cases in embodied and digital agents as complex tasks naturally admit to decomposition, hence bagged rewards.

**Audience:**

The findings of this paper are likely to interest a significant portion of TMLR's audience. The topic of RL with sparse or delayed rewards is highly relevant to real-world applications, e.g. robotics, and AI agents as AI agents such as DeepResearch increasingly tackle complex tasks. Furthermore, the proposed method's simplicity and potential for integration with existing RL frameworks make it appealing to researchers and practitioners in the field.

**Claims And Evidence:**

The claims made in the submission are supported by accurate, convincing, and clear evidence. The authors provide both theoretical and empirical analyses to substantiate their proposed method for reward redistribution in Reinforcement Learning from Bagged Rewards (RLBR). The theoretical foundation is well-articulated, particularly through the formulation of Bagged Reward Markov Decision Processes (BRMDPs) and the redistribution mechanism. Empirical results demonstrate consistent performance improvements over state-of-the-art methods across multiple experiments, with clear visualizations (e.g., Figure 4) supporting the claims. The authors also address reviewer concerns by clarifying assumptions and limitations in the revised manuscript.

---

> ### Author Response · Authors · 2025-04-21
>
> Dear Action Editor and Reviewers,
>
> Thank you very much for your helpful suggestions. We have revised the paper accordingly based on your feedback.
>
> We are grateful for your time, thoughtful feedback, and constructive guidance, which have greatly contributed to improving the quality of our submission.
>
> Best regards,
>
> Authors